# Placebo treatment affects brain systems related to affective and cognitive processes, but not nociceptive pain

Rotem Botvinik-Nezer [1,2,7] ✉, Bogdan Petre [2,7], Marta Ceko [3], Martin A. Lindquist[4], Naomi P. Friedman [5,6] & Tor D. Wager [2] ✉

Drug treatments for pain often do not outperform placebo, and a better understanding of placebo mechanisms is needed to improve treatment development and clinical practice. In a large-scale fMRI study ($N = 392$) with pre-registered analyses, we tested whether placebo analgesic treatment modulates nociceptive processes, and whether its effects generalize from conditioned to unconditioned pain modalities. Placebo treatment caused robust analgesia in conditioned thermal pain that generalized to unconditioned mechanical pain. However, placebo did not decrease pain-related fMRI activity in brain measures linked to nociceptive pain, including the Neurologic Pain Signature (NPS) and spinothalamic pathway regions, with strong support for null effects in Bayes Factor analyses. In addition, surprisingly, placebo increased activity in some spinothalamic regions for unconditioned mechanical pain. In contrast, placebo reduced activity in a neuromarker associated with higher-level contributions to pain, the Stimulus Intensity Independent Pain Signature (SIIPS), and affected activity in brain regions related to motivation and value, in both pain modalities. Individual differences in behavioral analgesia were correlated with neural changes in both modalities. Our results indicate that cognitive and affective processes primarily drive placebo analgesia, and show the potential of neuromarkers for separating treatment influences on nociception from influences on evaluative processes.

Throughout history, placebo effects have been variously considered as mysterious healing forces and tricks played upon the gullible by medical practitioners. Scientific research over the past decades has shown that placebo effects are neither of these. Rather, they are now understood to result from active, endogenous brain processes related to expectation, meaning, and predictive regulation of the body[1–4]. A substantial part of the benefit of many kinds of treatments—including conventional drug therapies[5–8], surgery[9,10], acupuncture[11,12], psychotherapy[13,14], and more—is related to these psychological and brain processes. The study of placebo effects is thus the study of the internal brain processes that promote health and healing. It is imperative to understand these processes more completely in order to harness them in clinical care. In particular, it is still unknown at which level sensory experiences and physiological responses are affected by these internal processes.

[1]Department of Psychology, The Hebrew University of Jerusalem, Jerusalem, Israel. [2]Department of Psychological and Brain Sciences, Dartmouth College, Hanover, NH, USA. [3]Institute of Cognitive Science, University of Colorado Boulder, Boulder, CO, USA. [4]Department of Biostatistics, Johns Hopkins University, Baltimore, MD, USA. [5]Institute for Behavioral Genetics, University of Colorado Boulder, Boulder, CO, USA. [6]Department of Psychology and Neuroscience, University of Colorado Boulder, Boulder, CO, USA. [7]These authors contributed equally: Rotem Botvinik-Nezer, Bogdan Petre. ✉e-mail: rotemb9@gmail.com; Tor.D.Wager@dartmouth.edu

Effective placebo treatments are thought to work by influencing 'meaning-making' systems[15–18]–internal models of the world that shape our interpretations of sensory events, including their underlying causes and their implications for the future. These models determine our predispositions to react negatively or positively to events (e.g., attentional biases, evaluative biases, and mindsets[19,20]), and facilitate enhanced or reduced reactions in perceptual and affective circuits[17]. These internal models provide predictive signals that are integrated with incoming sensory experience to produce experienced bodily sensations and symptoms[21–23]. For instance, current theories posit that pain is constructed by integrating 'top-down' context-based predictions with afferent nociceptive signals according to principles of Bayesian inference[24,25]. Prediction errors (discrepancies between sensory input and predicted values) are propagated upstream for learning (refining the internal model). In this way, placebo treatments–accompanied by suggestions, social cues, and prior experiences of success–can affect the neural construction of pain[24,26–33]. Similar accounts have been increasingly used to explain symptoms and cognitive distortions in multiple disorders[34–38].

However, a critical open question remains: How fundamentally do placebos and other manipulations of predictive models affect sensory processing pathways? This question is important because its answer is the difference between a profound analgesic effect on responses (and plasticity) throughout the nervous system, effects on higher-level construction of pain experience, or transient biases in decision-making. By some accounts, predictions can propagate down multiple levels of perceptual hierarchies to affect sensory processing at the earliest stages–e.g., visual responses in visual thalamic pathways and V1[23,39] and nociceptive processing in the spinal cord[24]. In support of this view, early studies of placebo analgesia provided evidence for placebo or nocebo (aversive predictions) effects on the spinal cord[40,41], release of endogenous opioids[42–46], increases in putative descending pain-control pathways in the brainstem[43,47,48], and decreased pain-related activity in the spino-thalamic tract[49,50]. Meta- and mega-analyses of placebo analgesia[51–54] have found reductions in areas associated with pain processing, including anterior midcingulate (aMCC), medial and ventrolateral thalamus, and anterior and (less consistently) dorsal posterior insula (aIns and dpIns). Reductions in these areas correlate with behavioral placebo analgesia[53,54].

However, some evidence suggest that many of these placebo-induced reductions in 'pain processing' may be related to affective and decision-making processes rather than nociception, implying a later stage of influence. aMCC and aIns signals are influenced by a wide variety of processes, from emotion to language to motor control[55–57]. dpIns is the cortical area most selective for nociception[58–60], but it also responds to non-somatic, emotional stimuli in some cases[60,61]. Thus, placebo influences on these regions do not necessarily entail effects on nociception.

Stronger tests can be provided by multivariate brain signatures, or neuromarkers[62], that can track noxious stimulus intensity and pain with substantially higher sensitivity (larger effect sizes) and specificity than individual regions[63–65]. In particular, the neurologic pain signature (NPS)[66] tracks the intensity of nociceptive input and predicts reported pain with very large effect sizes across cohorts (e.g., $d = 1.45$[67] or 2.3[54])–and is also largely specific to nociceptive pain: It does not respond to multiple types of non-nociceptive affective stimuli[65,68–70]. Tests of placebo effects on the NPS in individual studies[71] and a recent participant-level meta-analysis across 20 studies (603 participants)[54] show little influence of placebo effects. While a significant reduction with placebo was found, its effect size was very small ($d = 0.08$) compared with the robust behavioral placebo analgesia ($d = 0.66$), paralleling earlier findings[72] showing that neural placebo effects on noxious stimulus-evoked electrophysiological potentials were significant but too small to explain the behavioral analgesic effects of placebo.

Neuromarkers can also reduce the complexity of multiple testing, providing increased power to address questions about the correlates of individual differences in placebo effects. For example, recent studies have shown that the magnitude of placebo (or nocebo) effects might depend on sex[73–75], race[76], learning patterns[77], and baseline pain intensity[78] and variability[79–81], among other factors. Factors like sex might interact with specific characteristics of the placebo induction, such as whether conditioning is used to reinforce expectations[73–75]. Examining such interactions in a large number of brain regions with unknown direct relevance for analgesia is complex, and neuromarkers can simplify the space of tests by focusing on a few brain measures with stronger measurement properties and direct relevance to the outcome (pain). In addition, a limitation of meta-analyses in this respect is that they average small-sample effects from diverse paradigms. Large-sample studies such as the present one can inform on neural and behavioral placebo effects in a uniform experimental context, avoiding the averaging of heterogeneous effects unavoidable in meta-analyses.

Definitive tests of placebo effects on nociceptive and extra-nociceptive pain-related brain processes require highly powered tests of both positive (significant) and null effects (e.g., using Bayes factors[82]) in large samples, and the assessment of effect sizes in a priori markers. Here, we report such tests on the largest single neuroimaging sample of placebo analgesia to date ($N = 392$ after exclusions). The sample included participants from the Colorado Community Twin Sample, with monozygotic and dizygotic twins. Dependencies at the family level (i.e., within monozygotic and dizygotic twin pairs) were controlled with a mixed effects model design (see Methods). We tested placebo effects on thermal pain reinforced by a placebo response conditioning procedure[83] and transfer to unreinforced effects on mechanical pain (Fig. 1 and Methods). Signature-based analyses focused on the NPS and a second neuromarker for higher-level endogenous contributions to pain–the stimulus intensity independent pain signature (SIIPS[84])–originally trained and validated across six studies to predict pain ratings after removing variance associated with stimulus intensity and the NPS. Unlike NPS, SIIPS has been found to mediate the effects of several psychological manipulations on pain, including effects of conditioned auditory cues, visual cues, and perceived control[84]. In addition, we tested three sets of pre-registered brain regions of interest (ROIs; Supplementary Table 1; for pre-registration see https://osf.io/unh7f and "Methods" section). One set focused on regions most closely associated with nociceptive pain, the second one focused on sub-regions of the SIIPS signature, and the third set focused on prefrontal and striatal regions broadly involved in motivation, value, attention, and emotion regulation[50,51,85–88].

In addition to tests of nociceptive and affective pain neuro-markers and regions, this study addressed an important, unanswered question about transfer (i.e., generalization) of placebo effects. Historical studies have found that placebo effects do not generalize across different types of pain (e.g., labor, postpartum, and experimental, ischemic muscle pain[89]), and minor variations in context[90]. Moreover, substantial attempts to identify "placebo responders" across domains have been mostly unsuccessful[91]. On the other hand, one of the main underlying mechanisms of placebo effects is associative learning, which is known to generalize across stimuli following successful learning[92–94]. Furthermore, it has been shown that placebo effects transfer across different routes of drug administration[95] and over time based on treatment history[96,97], and also across domains in some cases (e.g., from pain to negative emotions[98] or from pain to motor performance[99]). In many cases, rather than across domains, placebo effects may transfer between modalities and stimuli within the same domain (e.g., between thermal and mechanical pain, but not from pain to itch[100]). No or weak placebo effects in the transfer condition would indicate the involvement of specific learning mechanisms in placebo analgesia, whereas strong effects would suggest involvement of inferential and expectancy-related processes[101–103].

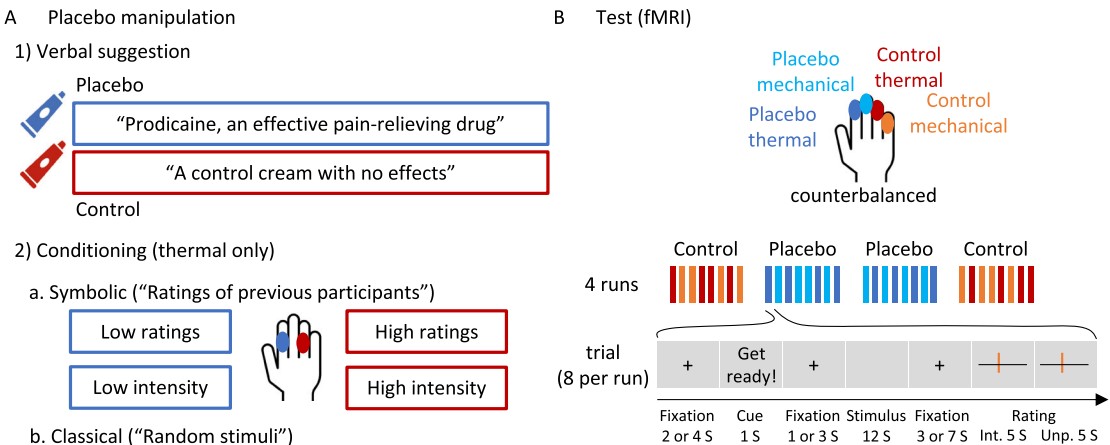

A Placebo manipulation
1) Verbal suggestion

B Test (fMRI)

**Fig. 1 | Experimental design. A** First, participants were introduced to the same cream, once presented as "Prodicaine, an effective pain-relieving drug" and once presented as "a control cream with no effects". Then they went through two conditioning phases: (1) A symbolic conditioning phase, in which "ratings of previous participants" were presented for a series of sham pain trials, with ratings systematically higher for the control compared to the placebo skin site. (2) A classical conditioning phase, in which participants experienced a series of thermal stimuli that they thought were random but in fact were experimentally manipulated to be higher for the control compared to the placebo skin site. **B** Then, the test task took place in the MRI scanner. The task consisted of four runs (order: control, placebo, placebo, control), each including eight trials, four with thermal stimulation and four with mechanical (not conditioned) stimulation. Stimuli in the test task were from three intensity levels per modality. In each trial, participants saw a cue, then experienced the stimulus and rated its intensity and unpleasantness. For further details see "Methods" section. S seconds, Int. intensity, Unp. unpleasantness.

## Results

### Behavioral placebo analgesia

First, we tested the effect of the placebo manipulation on pain ratings. We included three levels of intensity for each modality (thermal and mechanical pain), selected from pilot testing to be painful and tolerable in a broad population sample (thermal: 46.5, 47, and 47.5 °C; mechanical: 6, 7, and 8 kg/cm²). This allowed us to test for stimulus intensity effects, placebo effects, and their interaction on all outcomes, with a mixed-effects model controlling for the familial structure (see "Methods" section). Subjective ratings of stimulus intensity and unpleasantness were highly correlated across trials within participants (*Pearson's r* was computed for each of 372 participants: across participants, median = 0.956, *M* = 0.896, *sd* = 0.171). Therefore, we focused on intensity ratings, which are typically less responsive than unpleasantness ratings to psychological interventions[104]. The same pattern of effects was found with the unpleasantness ratings (see Supplementary Information). The ratings were provided on a Labeled Magnitude Scale (LMS[105,106]; 0 = no pain / not at all; 0.014 = barely detectable; 0.061 = weak; 0.172 = moderate; 0.354 = strong; 0.533 = very strong; 1 = strongest / worst pain imaginable).

**Behavioral ratings: Thermal pain.** As shown in Fig. 2, thermal pain ratings increased with stimulus intensity (*M* [averaged across conditions] = 0.129, 0.148, and 0.191 for low, medium, and high intensity; Intensity effect: $\beta = 0.409$, SE = 0.031, $t_{(416.2)} = 13.32$, $p < 0.001$, 95% CI = [0.348, 0.469]) and were lower in the Placebo compared to the Control condition (Placebo *M* = 0.129, Control *M* = 0.183; Placebo effect: $\beta = -0.359$, SE = 0.037, $t_{(230.9)} = -9.73$, $p < 0.001$, 95% CI = [−0.432, −0.286], $d = 0.53$). The Intensity x Placebo interaction was not significant ($\beta = -0.080$, SE = 0.053, $t_{(1282.9)} = -1.51$, $p = 0.131$, 95% CI = [−0.183, 0.024]). These results were robust to the inclusion of demographic covariates (sex and age, see "Robustness to covariates and demographic effects" in the Supplementary Information). In addition, individual differences in placebo analgesia were significantly correlated with pre-scan ratings of expected Prodicaine efficacy ($\beta = 0.107$, SE = 0.036, $t_{(339)} = 3.00$, $p = 0.003$, 95% CI = [0.368, 0.176]), indicating that participants who expected higher efficacy experienced stronger placebo analgesia.

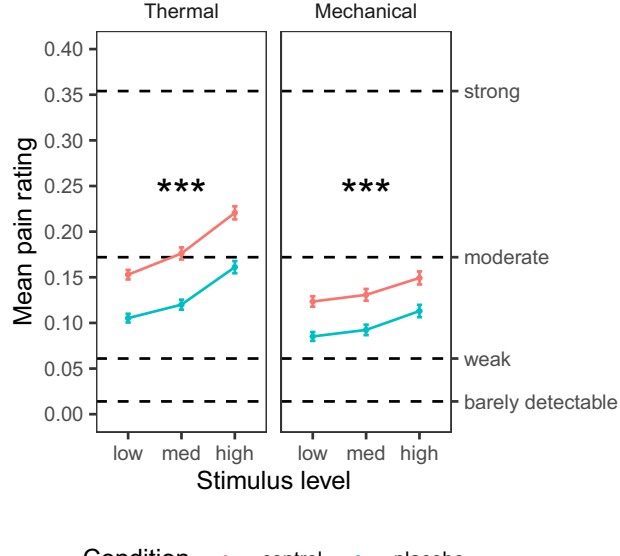

**Fig. 2 | Behavioral results.** Mean pain ratings for Control (red) and Placebo (blue) condition, for each combination of modality and stimulus level (*N* = 374 participants). Error bars represent within-participant standard error of the mean, based on Morey, 2008[107]. Asterisks represent significance of the placebo effect (mixed-effects model, uncorrected): *$p < 0.05$, **$p < 0.01$, ***$p < 0.001$. For additional visualizations see Supplementary Fig. 1.

**Behavioral ratings: Mechanical pain.** Although the placebo cream (Prodicaine) was only conditioned with the thermal stimuli, its effect transferred from the conditioned thermal modality to the unconditioned mechanical modality. Mechanical pain ratings increased with Stimulus Intensity (*M* = 0.104, 0.112, and 0.131; Intensity effect: $\beta = 0.202$, SE = 0.032, $t_{(121.7)} = 6.39$, $p < 0.001$, 95% CI = [0.139, 0.264]) and decreased with Placebo (Placebo *M* = 0.097, Control *M* = 0.134; Placebo effect: $\beta = -0.244$, SE = 0.039, $t_{(345.1)} = -6.24$, $p < 0.001$, 95% CI = [−0.321, −0.167], $d = 0.34$), with no significant interaction ($\beta = 0.013$, SE = 0.038, $t_{(1089.2)} = 0.33$, $p = 0.741$, 95% CI = [−0.062,

0.087]). These results were robust to the inclusion of demographic covariates (Supplementary Information, "Robustness to covariates and demographic effects"). Pre-scan expectations of Prodicaine efficacy were not correlated with the placebo effect in the mechanical modality ($\beta = 0.010$, SE = 0.044, $t_{(163.5)} = 0.23$, $p = 0.817$, 95% CI = [−0.077, 0.097]).

Furthermore, there was no significant difference in the placebo effect between the thermal and mechanical modalities ($\beta = -0.005$, SE = 0.051, $t_{(202)} = -0.09$, $p = 0.925$, 95% CI = [−0.106, 0.096]). Placebo effects were positively correlated across thermal and mechanical modalities, indicating that participants with stronger placebo effects in the thermal condition also showed a stronger placebo effect in the mechanical modality ($\beta = 0.215$, SE = 0.071, $t_{(151.2)} = 3.03$, $p = 0.003$, 95% CI = [0.075, 0.355]), providing additional evidence for stable placebo responses across pain types and conditioned vs. transfer modalities.

### Placebo effects on nociceptive processes

**Placebo effects on the Neurologic Pain Signature.** The NPS (Fig. 3A[66]) served as a neuromarker for nociceptive pain-related processes based on previous work. As described in the introduction, the NPS is highly sensitive and specific to nociceptive pain.

**NPS: Thermal pain.** As shown in Fig. 3B, replicating previous findings, the NPS score was positive during thermal stimuli ($d = 1.11$), and increased with increasing stimulus intensity ($\beta = 0.245$, SE = 0.041, $t_{(644.5)} = 6.00$, $p < 0.001$, 95% CI = [0.165, 0.325]). However, the NPS response was not significantly affected by placebo treatment ($\beta = -0.030$, SE = 0.036, $t_{(225.9)} = -0.83$, $p = 0.408$, 95% CI = [−0.101, 0.041], $d = 0.04$), and the Intensity x Placebo interaction was not significant ($\beta = 0.074$, SE = 0.078, $t_{(1351.5)} = 0.95$, $p = 0.342$, 95% CI = [−0.079, 0.227]). A Bayes Factor (BF) analysis revealed strong evidence in favor of the null hypothesis of no placebo effect (BF = 0.044, 23:1 odds in favor of the null, proportional error estimate = 9.80%). Since there is some disagreement on certain analytical choices in the context of Bayes Factor analysis[108], we further explored the robustness of these results to different analytical choices, particularly with regard to the width of the prior distribution and inclusion of interaction terms. Such variations did not change the conclusion, with all models indicating strong to extreme evidence in favor of the null (BF range 0.036-0.069; Supplementary Table 2). Furthermore, pre-scan expectations of Prodicaine efficacy were not correlated with the placebo effect on the NPS score ($\beta = 0.033$, SE = 0.034, $t_{(230.8)} = 0.99$, $p = 0.323$, 95% CI = [−0.033, 0.100]).

**NPS: Mechanical pain.** The NPS score was positive during mechanical pain ($d = 1.02$), and increased with increased stimulus intensity ($\beta = 0.164$, SE = 0.041, $t_{(804.1)} = 4.02$, $p < 0.001$, 95% CI = [0.084, 0.244]), but was not affected by placebo treatment ($\beta = 0.008$, SE = 0.037, $t_{(308.7)} = 0.23$, $p = 0.822$, 95% CI = [−0.064, 0.081], $d = -0.02$; Fig. 3B), and the Intensity x Placebo interaction was not significant ($\beta = 0.132$, SE = 0.079, $t_{(1529)} = 1.67$, $p = 0.095$, 95% CI = [−0.023, 0.287]). Bayes Factor analysis provided strong evidence against the presence of a placebo effect on the NPS response (BF = 0.036, 28:1 odds in favor of the null, proportional error estimate = 11.7%), and this result was robust to analytical variations with strong to extreme evidence across models (BF range 0.001 − 0.049; Supplementary Table 2). As in the thermal modality, pre-scan expectations of Prodicaine efficacy were not correlated with the placebo effect on the NPS score in mechanical trials ($\beta = 0.012$, SE = 0.033, $t_{(351.8)} = 0.38$, $p = 0.703$, 95% CI = [−0.051, 0.076]).

Together, these analyses indicated that placebo does not affect fMRI activity in the most widely validated neuromarker to date for nociceptive pain. These results were robust to the inclusion of demographic covariates (Supplementary Information, "Robustness to covariates and demographic effects").

**A priori nociceptive regions of interest.** The NPS is only one measure and does not capture all aspects of pain processing. Thus, to further assess effects on regions associated with nociception, we tested effects of placebo, stimulus intensity, and their interaction in a set of seven a priori brain regions associated with nociception (six of which were pre-registered, see "Methods" section; for full statistics see Tables 1 and 2). Because false negatives and false positives here were equally important, we report results based on $p < 0.05$ without correcting for multiple comparisons across regions. This is because of the nature of the current study, focusing largely on testing pre-registered neural signatures and regions identified in previous literature regions with a substantially larger sample size. Nevertheless, to be slightly more conservative, we note when a result does not survive Bonferroni correction within its set of regions (e.g., within the set of seven nociceptive regions).

**Nociceptive ROIs: Thermal pain.** Of these regions, the heat-evoked response increased with stimulus intensity in the anterior mid-cingulate cortex (aMCC), right (contralateral) dpIns, left (ipsilateral) dpIns and PAG (the effects in the PAG and left dpIns do not survive Bonferroni correction), but not in the right and left ventral posterior (VPL/M) or medial thalamus (Table 1). The placebo treatment did not significantly modulate the response to painful heat in any of these regions (all $p$s > 0.11, Table 2 and Fig. 3C). Bayes Factor analysis revealed strong evidence in favor of the null hypothesis (no effect of placebo) in all these a priori brain nociceptive regions, except the left VPL/M thalamus, which showed moderate evidence for the null (Supplementary Table 2). Again, these results were robust to the inclusion of interaction terms (leading to stronger evidence in favor of the null model) and width of the prior distribution (leading to slightly weaker evidence for some regions, yet still moderate to very strong evidence in favor of the null for all regions; all BFs < 0.143). In addition, Intensity x Placebo interactions were non-significant in all regions. These results provide strong evidence that placebo treatment does not modulate response to thermal stimuli in these a priori nociceptive brain regions.

**Nociceptive ROIs: Mechanical pain.** Pressure-evoked responses increased with stimulus intensity in the right (contralateral) dpIns, PAG and aMCC, but not in the left dpIns, right and left VPL/M thalamus, and medial thalamus (Table 1). Surprisingly, in the mechanical pain transfer condition, placebo treatment increased activity in all these a priori nociception-related regions compared with the control condition, except for the PAG ($p = 0.012$ or lower across individual regions; Table 2 and Fig. 3C). Bayes Factor analysis indicated extreme evidence in favor of the alternative hypothesis (increased activity during placebo compared to control) in the left and right dpIns, left VPL/M thalamus and medial thalamus, strong evidence in the right VPL/M thalamus, and moderate evidence in the aMCC (these results were robust to the width of the prior distribution, but not to the inclusion of an interaction term; Supplementary Table 2). In the PAG, the Bayes Factor analysis provided strong evidence in favor of the null hypothesis (no placebo effect). Intensity x Placebo interactions were all non-significant.

Since there were several surprising local increased activations for Placebo compared to Control during mechanical pain stimuli specifically, we performed exploratory analysis to test whether these effects can be accounted for by carry-over effects from the anticipation period. Anticipation responses did not account for these effects, as anticipatory brain activity in these regions did not significantly differ between Placebo and Control runs for the mechanical trials, except for the aMCC where anticipatory activity in mechanical trials was significantly lower in Placebo compared to Control ($\beta = -0.098$, SE = 0.049, $t_{(287.4)} = -2.00$, $p = 0.047$, 95% CI = [−0.195, −0.001]). A full analysis of the anticipation data is beyond the scope of the current paper and will be reported in future ones.

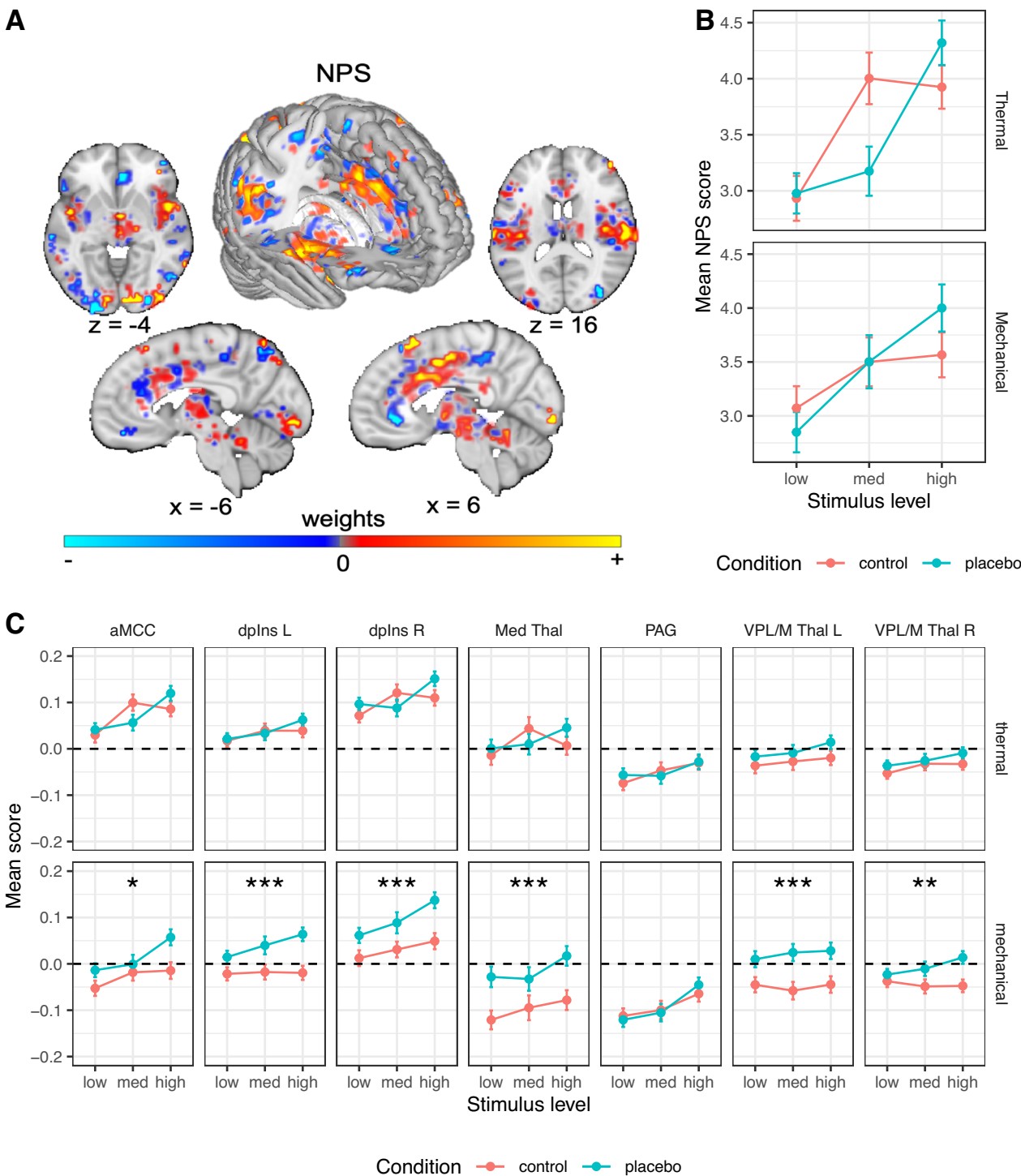

**Fig. 3 | Neural results: a priori nociceptive neuromarker and ROIs. A** The NPS, an fMRI measure optimized to predict pain intensity[66]. Dominant model parameters are highlighted by transparency scaling. **B** The mean NPS scores (based on dot product) across participants are presented, in the Control (red) and Placebo (blue) condition, for each combination of modality and stimulus level. For additional visualizations see Supplementary Fig. 2. **C** The mean signal across all voxels of each a priori nociceptive ROI across participants are presented, in the Control (red) and Placebo (blue) condition, for each combination of modality and stimulus level. In panels **B** and **C**, error bars represent within-participant standard error of the mean, based on Morey, 2008[107]. *N* = 392 participants. Asterisks represent significance of the placebo effect (mixed-effects model, uncorrected): *$p < 0.05$, **$p < 0.01$, ***$p < 0.001$. For additional visualizations see Supplementary Fig. 3. Regions' abbreviation: L left, R right, NPS Neurologic Pain Signature, aMCC anterior mid-cingulate cortex, dpIns dorsal posterior insula, Med Thal medial thalamus, VPL/M Thal ventral posterior thalamus, PAG periaqueductal gray.

**Table 1 | Statistics for a priori nociceptive ROIs: stimulus level effect**

| Region | Modality | Estimate | 95% CI | SE | t | DF | p value |
|---|---|---|---|---|---|---|---|
| aMCC | Thermal | 0.166 | [0.084, 0.249] | 0.042 | 3.97 | 763.9 | **<0.001** |
| | Mechanical | 0.128 | [0.049, 0.208] | 0.040 | 3.18 | 461.9 | **0.002** |
| dpIns L | Thermal | 0.091 | [0.013, 0.169] | 0.040 | 2.29 | 589.2 | **0.023** |
| | Mechanical | 0.067 | [−0.012, 0.147] | 0.041 | 1.66 | 750.9 | 0.098 |
| dpIns R | Thermal | 0.116 | [0.038, 0.195] | 0.040 | 2.90 | 966.5 | **0.004** |
| | Mechanical | 0.125 | [0.048, 0.203] | 0.039 | 3.18 | 871.8 | **0.002** |
| PAG | Thermal | 0.102 | [0.016, 0.189] | 0.044 | 2.33 | 1296.6 | **0.020** |
| | Mechanical | 0.157 | [0.070, 0.244] | 0.044 | 3.56 | 369.1 | **<0.001** |
| Med Thal | Thermal | 0.064 | [−0.016, 0.144] | 0.041 | 1.58 | 537.2 | 0.115 |
| | Mechanical | 0.077 | [−0.003, 0.158] | 0.041 | 1.90 | 498.0 | 0.058 |
| VPL/M Thal L | Thermal | 0.059 | [−0.016, 0.134] | 0.038 | 1.55 | 1056.0 | 0.121 |
| | Mechanical | 0.019 | [−0.062, 0.101] | 0.041 | 0.47 | 341.7 | 0.637 |
| VPL/M Thal R | Thermal | 0.076 | [−0.007, 0.159] | 0.042 | 1.81 | 675.7 | 0.072 |
| | Mechanical | 0.038 | [−0.042, 0.118] | 0.041 | 0.94 | 1076.3 | 0.350 |

Full statistics for the mixed effects models of the activity in each a priori ROI, for the stimulus level effect. Significant *p* values (uncorrected *p* < 0.05) are marked in bold.
*CI* confidence interval, *DF* degrees of freedom, *L* left, *R* right, *aMCC* anterior midcingulate cortex, *dpIns* dorsal posterior insula, *Med Thal* medial thalamus, *VPL/M Thal* ventral posterior thalamus, *PAG* periaqueductal gray.

**Table 2 | Statistics for a priori nociceptive ROIs: placebo effect**

| Region | Modality | Estimate | 95% CI | SE | t | DF | p value |
|---|---|---|---|---|---|---|---|
| aMCC | Thermal | 0.003 | [−0.068, 0.075] | 0.036 | 0.10 | 223.5 | 0.924 |
| | Mechanical | 0.095 | [0.021, 0.169] | 0.038 | 2.53 | 398.2 | **0.012** |
| dpIns L | Thermal | 0.018 | [−0.050, 0.086] | 0.035 | 0.52 | 246.1 | 0.602 |
| | Mechanical | 0.143 | [0.075, 0.212] | 0.035 | 4.11 | 498.8 | **<0.001** |
| dpIns R | Thermal | 0.027 | [−0.042, 0.096] | 0.035 | 0.77 | 226.7 | 0.444 |
| | Mechanical | 0.136 | [0.065, 0.207] | 0.036 | 3.78 | 381.2 | **<0.001** |
| PAG | Thermal | 0.01 | [−0.073, 0.094] | 0.042 | 0.24 | 213.9 | 0.809 |
| | Mechanical | −0.001 | [−0.083, 0.082] | 0.042 | −0.02 | 231.5 | 0.988 |
| Med Thal | Thermal | 0.016 | [−0.060, 0.091] | 0.038 | 0.41 | 364.1 | 0.685 |
| | Mechanical | 0.142 | [0.069, 0.215] | 0.037 | 3.81 | 472.3 | **<0.001** |
| VPL/M Thal L | Thermal | 0.061 | [−0.015, 0.136] | 0.038 | 1.59 | 223.9 | 0.114 |
| | Mechanical | 0.153 | [0.082, 0.225] | 0.036 | 4.20 | 329.3 | **<0.001** |
| VPL/M Thal R | Thermal | 0.047 | [−0.024, 0.119] | 0.036 | 1.31 | 199.5 | 0.193 |
| | Mechanical | 0.108 | [0.036, 0.180] | 0.037 | 2.93 | 389.4 | **0.004** |

Full statistics for the mixed effects models of the activity in each a priori ROI, for the placebo effect. Significant *p* values (uncorrected *p* < 0.05) are marked in bold. Negative *t*-values indicate reductions with Placebo vs. Control, which was the expected direction; positive *t*-values indicate paradoxical increases.
*CI* confidence interval, *DF* degrees of freedom, *L* left, *R* right, *aMCC* anterior midcingulate cortex, *dpIns* dorsal posterior insula, *Med Thal* medial thalamus, *VPL/M Thal* ventral posterior thalamus, *PAG* periaqueductal gray.

Taken together, these findings provide definitive evidence that placebo analgesia does not reduce activity in regions associated with nociceptive pain on average in this study.

### Placebo effects on higher-level processes

Pain perception is more than nociceptive processing, and includes higher-level cognitive and affective processes. Thus, we also tested the effect of placebo on such higher-level pain processing, including a priori neuromarker and regions of interest from previous studies.

**Placebo effects on the Stimulus Intensity-Independent Pain Signature.** The SIIPS[84] (Fig. 4A) is a neuromarker trained to predict pain independent of stimulus intensity and the NPS. SIIPS was designed to capture higher-level, endogenous brain influences on pain construction, beyond the nociceptive effects captured by the NPS. This signature was trained on four different datasets (*N* = 137 participants overall) and tested on two independent datasets (*N* = 46), explaining trial-level variance in pain ratings beyond the variance explained by the input intensity and NPS, and mediating the effects of three

psychological manipulations on pain (two eliciting expectancy effects and one manipulating perceived control). The SIIPS includes specific activity patterns in key placebo-linked regions, including vmPFC, dlPFC, hippocampus, and nucleus accumbens (NAc).

**SIIPS: Thermal pain.** As predicted, the SIIPS stimulus-evoked response was significantly reduced by the placebo treatment ($\beta = -0.129$, SE = 0.034, $t_{(417.4)} = -3.83$, $p < 0.001$, 95% CI = [−0.195, −0.063], $d = 0.19$), indicating that the endogenous processes captured by SIIPS are modulated by placebo effects (Fig. 4B). Bayes Factor analysis provided extreme evidence for this conclusion ($BF = 93.92$, proportional error = 13.9%; extreme evidence across variations beside inclusion of an interaction term; Supplementary Table 3).

Though the SIIPS was trained to explain variance in pain reports after controlling for stimulus intensity and nociceptive processes, it has been found to respond more to stronger stimuli in previous studies[84]. This can occur if variations in some sub-regions of the SIIPS that are considered nociceptive (e.g., parts of the insula and cingulate cortex) contribute to pain beyond intensity encoding. Indeed, the

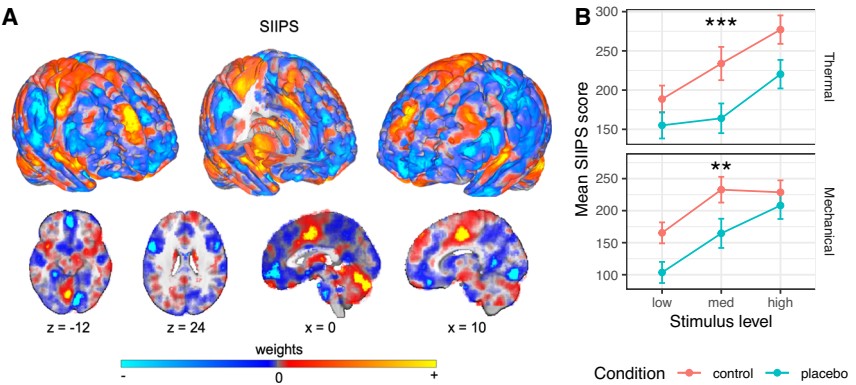

**Fig. 4 | Neural results: a priori higher-level pain processing neuromarker. A** The SIIPS signature, fMRI measure optimized to predict pain beyond nociception[84]. Dominant model parameters are highlighted by transparency scaling. **B** The mean SIIPS scores (based on dot product) across participants are presented, in the Control (red) and Placebo (blue) condition, for each combination of modality and stimulus level. Error bars represent within-participant standard error of the mean, based on Morey, 2008[107]. For additional visualizations see Supplementary Fig. 4. $N = 392$ participants. Asterisks represent significance of the placebo effect (mixed-effects model, uncorrected): *$p < 0.05$, **$p < 0.01$, ***$p < 0.001$. For visualizations of SIIPS subregions see Supplementary Figs. 5 and 6.

SIIPS stimulus-evoked response was also stronger for more intense stimuli in the current study ($\beta = 0.181$, SE = 0.038, $t_{(1408.3)} = 4.77$, $p < 0.001$, 95% CI = [0.106, 0.255]). The Intensity x Placebo interaction was non-significant ($\beta = -0.055$, SE = 0.075, $t_{(1559.9)} = -0.73$, $p = 0.467$, 95% CI = [−0.203, 0.093]). Pre-scan expectations of Prodicaine efficacy were not correlated with the placebo effect on the SIIPS score ($\beta = 0.042$, SE = 0.031, $t_{(251.4)} = 1.35$, $p = 0.178$, 95% CI = [−0.019, 0.103]).

Finally, the placebo-induced reduction in neuromarker score was significantly larger for the SIIPS compared to the NPS (estimate = 0.100, SE = 0.046, $t_{(678.81)} = 2.175$, $p = 0.030$, 95% CI = [0.010, 0.189]; note that the neuromarker scores were z-scored within each neuromarker and modality, to allow a direct comparison between these two outcomes that are measured on different scales; see Methods).

**SIIPS: Mechanical pain.** The effect of placebo on the SIIPS stimulus-evoked response transferred to the unconditioned mechanical pain modality, with lower SIIPS response in the Placebo compared to the Control treatment ($\beta = -0.112$, SE = 0.037, $t_{(286.1)} = -3.00$, $p = 0.003$, 95% CI = [−0.185, −0.039], $d = 0.15$; Fig. 4A). This effect was not significantly different from the placebo effect in the thermal modality ($\beta = 0.0002$, SE = 0.045, $t_{(213.5)} = 0.005$, $p = 0.996$, 95% CI = [−0.088, 0.088]). As in the thermal modality, the SIIPS response was also stronger for more intense mechanical stimuli ($\beta = 0.188$, SE = 0.039, $t_{(233)} = 4.75$, $p < 0.001$, 95% CI = [0.110, 0.265]). The Intensity × Placebo interaction was non-significant ($\beta = 0.090$, SE = 0.071, $t_{(1274.2)} = 1.27$, $p = 0.205$, 95% CI = [−0.049, −0.229]). The SIIPS results were robust to the inclusion of demographic covariates (Supplementary Information, "Robustness to covariates and demographic effects"). Like in the thermal modality, pre-scan expectations of Prodicaine efficacy were not correlated with the placebo effect on the SIIPS score in mechanical trials ($\beta = -0.021$, SE = 0.035, $t_{(299.3)} = -0.61$, $p = 0.539$, 95% CI = [−0.090, 0.047]). Moreover, the placebo-induced reduction in neuromarker score was again significantly larger for the SIIPS compared to the NPS (estimate = 0.124, SE = 0.045, $t_{(609.62)} = 2.766$, $p = 0.006$, 95% CI = [0.036, 0.211]).

**SIIPS: Subregions.** SIIPS is a multivariate signature that includes positive and negative weights across the brain. A subset of regions in the brain have robust pattern weights (i.e., they made consistent contributions to prediction across participants and studies in the original paper[84]). These subregions can be divided into three types: (1) Regions with positive weights (i.e., increased activity is associated with more pain) and which are established as targets of nociceptive inputs. These regions include the insula, thalamus, and cingulate cortex. These regions overlap with some of the gross anatomical regions included in the NPS. However, they reflect distinct local regions, and their weights in the SIIPS are not correlated with their weights in the NPS. Thus, they likely encode pain beyond the nociceptive processes captured by NPS. (2) Regions with positive weights and which are not known to be nociceptive, including the dmPFC and caudate. These regions likely encode pain processing beyond nociception. (3) Regions with negative weights (i.e., increased activity in these regions was associated with less pain), including areas in the NAc and the vmPFC. These regions are thought to encode cognitive and affective processes related to pain regulation and other pain-opposing processes, and are thought to play an important role in chronic pain[109–111].

We focused on eight specific subregions of SIIPS that showed non-significant responses to noxious stimulus intensity in the SIIPS' training studies and are thus considered non-nociceptive contributors to the pain experience. These subregions include two regions with positive weights (dmPFC and right middle temporal gyrus, rMTG) and six regions with negative weights (right lingual gyrus, left superior temporal gyrus [lSTG], left NAc, right temporal pole, left inferior temporal gyrus [lITG] and middle precentral gyrus). In addition to these eight pre-registered subregions of SIIPS, we also tested for effects in seven subregions that were not pre-registered but are of interest based on recent work, including the vmPFC, right nucleus accumbens, left and right dlPFC, right secondary somatosensory cortex, right sensorimotor cortex and left precuneus (for the results in these regions see Supplementary Information, "Additional, non-pre-registered SIIPS subregions").

For full statistics in all eight pre-registered subregions for both modalities, see Supplementary Table 4 and Supplementary Table 5. We report local pattern response in each region, which is the dot product of the activity in each voxel and its corresponding SIIPS pattern weights. The interpretation is somewhat different from the standard average activity reported for ROIs. In regions with positive weights, more positive pattern responses indicate greater activation and increased pain. In subregions with negative weights, positive pattern responses indicate deactivation and predict increased pain (activity in these regions is also scaled based on the weights across the different voxels). Thus, more positive responses indicated more pain-related activity in all regions, and we expected decreased pattern responses during Placebo compared to Control treatment (negative t-values) in all subregions. In "SIIPS-Pos" regions, decreased pattern response corresponds to decreased overall activity with placebo. In "SIIPS-Neg"

**Table 3 | Statistics for a priori ROIs for higher-level pain processing: stimulus level effect**

| Region | Modality | Estimate | 95% CI | SE | t | DF | p value |
|---|---|---|---|---|---|---|---|
| dlPFC L 1 | Thermal | −0.008 | [−0.089, 0.072] | 0.041 | −0.20 | 594.6 | 0.840 |
| | Mechanical | 0.017 | [−0.065, 0.098] | 0.042 | 0.40 | 1242.5 | 0.689 |
| dlPFC L 2 | Thermal | 0.077 | [−0.005, 0.159] | 0.042 | 1.85 | 758.5 | 0.065 |
| | Mechanical | 0.088 | [0.008, 0.169] | 0.041 | 2.16 | 487.3 | **0.031** |
| dlPFC R | Thermal | −0.001 | [−0.084, 0.081] | 0.042 | −0.04 | 553.6 | 0.972 |
| | Mechanical | −0.019 | [−0.103, 0.065] | 0.043 | −0.44 | 462.4 | 0.658 |
| Lateral OFC | Thermal | −0.027 | [−0.107, 0.053] | 0.041 | −0.67 | 616.0 | 0.505 |
| | Mechanical | −0.040 | [−0.122, 0.043] | 0.042 | −0.95 | 929.8 | 0.345 |
| Anterior OFC 1 | Thermal | −0.052 | [−0.131, 0.027] | 0.040 | −1.30 | 539.2 | 0.194 |
| | Mechanical | −0.018 | [−0.103, 0.067] | 0.043 | −0.42 | 1246.8 | 0.674 |
| Anterior OFC 2 | Thermal | −0.002 | [−0.086, 0.081] | 0.043 | −0.05 | 689.1 | 0.961 |
| | Mechanical | 0.039 | [−0.045, 0.123] | 0.043 | 0.91 | 657.2 | 0.362 |
| Mid-lateral OFC L | Thermal | −0.020 | [−0.098, 0.059] | 0.040 | −0.49 | 1132.8 | 0.623 |
| | Mechanical | 0.02 | [−0.069, 0.110] | 0.046 | 0.45 | 628.7 | 0.655 |
| Mid-lateral OFC R | Thermal | −0.003 | [−0.083, 0.077] | 0.041 | −0.07 | 647.5 | 0.948 |
| | Mechanical | 0.045 | [−0.041, 0.132] | 0.044 | 1.03 | 583.3 | 0.303 |
| NAc L | Thermal | −0.011 | [−0.094, 0.071] | 0.042 | −0.27 | 1370.7 | 0.785 |
| | Mechanical | 0.123 | [0.038, 0.208] | 0.043 | 2.84 | 1211.7 | **0.005** |
| NAc R | Thermal | 0.074 | [−0.004, 0.152] | 0.040 | 1.85 | 660.9 | 0.064 |
| | Mechanical | 0.071 | [−0.013, 0.154] | 0.042 | 1.67 | 647.6 | 0.096 |

Full statistics for the mixed effects models of the activity in each a priori higher-level pain processing ROI, for the stimulus level effect. Significant p values (uncorrected p < 0.05) are marked in bold.
CI confidence interval, DF degrees of freedom, L left, R right, dlPFC dorsolateral prefrontal cortex, OFC orbitofrontal cortex, NAc nucleus accumbens.

regions (e.g., NAc), decreased pattern response corresponds to increased overall activity with placebo.

dmPFC and rMTG (the regions with positive weights) showed no placebo effects in either thermal or mechanical pain (Supplementary Fig. 5). The only region with significant placebo-related decreases in local SIIPS pattern response during thermal pain was lSTG (however, this result did not survive Bonferroni correction). On mechanical pain trials, we found significant placebo-induced decreases in the right lingual gyrus, lSTG, left NAc, middle precentral gyrus and right temporal pole, but not lITG (and the effect in the right temporal pole did not survive Bonferroni correction). Unexpectedly, there was a significant negative effect of stimulus level in the rMTG in the thermal modality, and a significant positive effect of stimulus level in the dmPFC in the mechanical modality, but both of these effects did not survive Bonferroni correction.

**A priori higher-level processing regions of interest.** We tested ten a priori brain regions where increased response to noxious stimuli in placebo compared to control was found in previous studies. These regions are thus suggested to reflect higher-level processes that modulate the placebo effect, such as motivation, emotion regulation, and decision-making. These regions included the right dlPFC, two different areas of the left dlPFC (one more anterior than the other), two areas of the anterior orbitofrontal cortex (OFC; one more inferior and one more superior), lateral OFC, right and left middle lateral OFC, and right and left NAc (here, a simple region average, in contrast to the local pattern responses reported for SIIPS subregions). For full statistics for each individual region, see Tables 3 and 4.

**Higher-level ROIs: Thermal pain.** We found significant placebo-induced increases in the response to thermal stimuli in the more posterior of the two left dlPFC regions, the right dlPFC, the more superior of the two anterior OFC regions, and the lateral OFC (but only the left dlPFC survives Bonferroni correction; Table 4 and Fig. 5). There were no significant placebo effects in the right and left NAc, and in the second (more anterior) a priori area of the left dlPFC, the second (more

inferior) a priori area of the anterior OFC, and the right and left mid-lateral OFC.

**Higher-level ROIs: Mechanical pain.** During mechanical pain trials, we found significant placebo-induced increases in the more posterior region of the left dlPFC and the more superior region of the anterior OFC (as in thermal pain, though the latter result does not survive Bonferroni correction) and in the left and right NAc (Table 4 and Fig. 5). No significant placebo effects were found in the other a priori regions.

**Individual differences in placebo effects.** In the current paper, we have focused primarily on the causal effects of placebo treatment in experimental settings on pain at the group level, as reported above. Nevertheless, we also tested the correlations between individual differences in placebo analgesia and neural placebo-induced changes, as was done in previous studies. Importantly, correlations in small samples (as in most previous placebo fMRI studies) are unreliable, and meta-analyses cannot address this issue because of heterogeneity across studies. Thus, this is one of the first studies that could adequately test these behavioral-brain correlations. Nevertheless, such correlations do not imply causal effects of placebo, and can be driven by other processes (see Discussion).

**NPS.** Although there was no effect of placebo treatment on the NPS on average at the group level, we found that stronger placebo-induced NPS reductions were associated with stronger behavioral analgesia in both the thermal and mechanical modalities (mixed-effects model, see methods for details; thermal: $\beta = 0.198$, SE = 0.034, $t_{(106.2)} = 5.83$, $p < 0.001$, 95% CI = [0.131, 0.265]; mechanical: $\beta = 0.202$, SE = 0.037, $t_{(63)} = 5.46$, $p < 0.001$, 95% CI = [0.128, 0.276]; Fig. 6A). Since different participants use the rating scale differently, increasing skewness in the data, we tested the robustness of these results by testing Spearman's correlations based on the rank-order of the behavioral and neural Placebo - Control values across participants (ranked separately within each combination of pain modality and stimulus level). The correlation was significant with this nonparametric correlation test as well

**Table 4 | Statistics for a priori ROIs for higher-level pain processing: placebo effect**

| Region | Modality | Estimate | 95% CI | SE | t | DF | p value |
|---|---|---|---|---|---|---|---|
| dlPFC L 1 | Thermal | 0.105 | [0.035, 0.176] | 0.036 | 2.95 | 408.8 | **0.003** |
| | Mechanical | 0.114 | [0.040, 0.187] | 0.037 | 3.05 | 395.0 | **0.002** |
| dlPFC L 2 | Thermal | 0.032 | [−0.044, 0.108] | 0.039 | 0.82 | 229.2 | 0.413 |
| | Mechanical | 0.068 | [−0.008, 0.144] | 0.038 | 1.77 | 321.5 | 0.077 |
| dlPFC R | Thermal | 0.083 | [0.014, 0.151] | 0.035 | 2.37 | 473.7 | **0.018** |
| | Mechanical | 0.049 | [−0.027, 0.125] | 0.039 | 1.27 | 260.8 | 0.207 |
| Lateral OFC | Thermal | 0.076 | [0.010, 0.142] | 0.034 | 2.26 | 362.3 | **0.024** |
| | Mechanical | 0.014 | [−0.060, 0.088] | 0.038 | 0.36 | 500.5 | 0.717 |
| Anterior OFC 1 | Thermal | 0.072 | [−0.001, 0.145] | 0.037 | 1.94 | 246.5 | 0.054 |
| | Mechanical | 0.024 | [−0.051, 0.098] | 0.038 | 0.62 | 405.5 | 0.534 |
| Anterior OFC 2 | Thermal | 0.081 | [0.007, 0.155] | 0.038 | 2.14 | 373.3 | **0.033** |
| | Mechanical | 0.106 | [0.026, 0.187] | 0.041 | 2.60 | 224.2 | **0.010** |
| Mid-lateral OFC L | Thermal | 0.062 | [−0.015, 0.140] | 0.039 | 1.58 | 321.3 | 0.115 |
| | Mechanical | 0.044 | [−0.029, 0.118] | 0.037 | 1.18 | 748.7 | 0.238 |
| Mid-lateral OFC R | Thermal | 0.069 | [−0.007, 0.145] | 0.039 | 1.79 | 360.6 | 0.075 |
| | Mechanical | 0.048 | [−0.026, 0.121] | 0.037 | 1.28 | 765.3 | 0.201 |
| NAc L | Thermal | 0.001 | [−0.078, 0.079] | 0.040 | 0.02 | 280.3 | 0.987 |
| | Mechanical | 0.156 | [0.080, 0.233] | 0.039 | 4.02 | 434.3 | **<0.001** |
| NAc R | Thermal | 0.014 | [−0.061, 0.090] | 0.038 | 0.37 | 376.5 | 0.712 |
| | Mechanical | 0.178 | [0.105, 0.251] | 0.037 | 4.78 | 370.2 | **<0.001** |

Full statistics for the mixed effects models of the activity in each a priori higher-level pain processing ROI, for the placebo effect. Significant p values (uncorrected p < 0.05) are marked in bold.
Significant p values (uncorrected p < 0.05) are marked in bold.
*CI* confidence interval, *DF* degrees of freedom, *L* left, *R* right, *dlPFC* dorsolateral prefrontal cortex, *OFC* orbitofrontal cortex, *NAc* nucleus accumbens.

(thermal: $\beta$ = 0.158, SE = 0.033, $t_{(159.8)}$ = 4.84, $p$ < 0.001, 95% CI = [0.094, 0.222]; mechanical: $\beta$ = 0.140, SE = 0.032, $t_{(258.4)}$ = 4.43, $p$ < 0.001, 95% CI = [0.078. 0.202]). Since there was no group effect of placebo treatment on the NPS, these findings could result from NPS reductions in stronger placebo responders, or from other correlated factors such as random differences in sensitivity between the Placebo and Control skin sites. We cannot identify placebo responders from independent data in the current study, and therefore we cannot dissociate between these two alternatives (see Discussion). NPS placebo-induced reductions were not correlated between the thermal and mechanical modalities ($\beta$ = 0.067, SE = 0.056, $t_{(99.3)}$ = 1.18, $p$ = 0.239, 95% CI = [−0.045, 0.179]).

**Nociceptive ROIs.** We tested the correlation between the placebo-induced neural reductions and the behavioral analgesia in each of the a priori nociceptive ROIs (see Supplementary Table 8 for full statistics). In the thermal modality, the correlation was significantly positive in the aMCC, right dpIns, PAG and medial thalamus (the result in the medial thalamus did not survive Bonferroni correction). In the mechanical modality, we found significant positive correlations in the aMCC, right and left dpIns, and PAG. These indicate greater placebo-induced reduction in brain response with greater analgesia, as expected. Although the remaining regions did not show statistically significant differences, trends were all in the positive direction in both modalities.

**SIIPS.** Placebo-induced neural reductions in SIIPS scores were significantly correlated with behavioral analgesia in both modalities, whether measured linearly (thermal: $\beta$ = 0.229, SE = 0.033, $t_{(302.3)}$ = 7.00, $p$ < 0.001, 95% CI = [0.165, 0.294]; mechanical: $\beta$ = 0.241, SE = 0.036, $t_{(67.9)}$ = 6.78, $p$ < 0.001, 95% CI = [0.170, 0.312]; Fig. 6B) or based on ranks (Spearman; thermal: $\beta$ = 0.207, SE = 0.032, $t_{(170.3)}$ = 6.47, $p$ < 0.001, 95% CI = [0.144, 0.270]; mechanical: $\beta$ = 0.181, SE = 0.034, $t_{(177)}$ = 5.27, $p$ < 0.001, 95% CI = [0.113, 0.249]). Moreover, placebo-induced SIIPS reductions positively correlated with placebo-induced NPS reductions in both modalities (thermal: $\beta$ = 0.243,

SE = 0.033, $t_{(243.1)}$ = 7.32, $p$ < 0.001, 95% CI = [0.178, 0.309]; mechanical: $\beta$ = 0.188, SE = 0.036, $t_{(173.5)}$ = 5.21, $p$ < 0.001, 95% CI = [0.117, 0.260]). Placebo-induced SIIPS reductions did not correlate across thermal and mechanical modalities ($\beta$ = 0.081, SE = 0.065, $t_{(85.4)}$ = 1.26, $p$ = 0.212, 95% CI = [−0.047, 0.210]).

**SIIPS subregions.** The correlation between the behavioral and neural placebo-induced reductions in local pattern responses within SIIPS subregions was significant only for the right lingual gyrus (a positive correlation, as expected) in the thermal modality, and the left superior temporal gyrus and right temporal pole (both with negative correlations, surprisingly, however not surviving correction for multiple comparisons) in the mechanical modality (see Supplementary Table 9 for full statistics).

**Higher level ROIs.** In these a priori regions, we expected greater placebo-induced responses (more negative control - placebo scores) to predict stronger behavioral analgesia (control - placebo; a negative correlation). Nevertheless, we did not find significant negative correlations in any of these regions in both modalities. Conversely, in the thermal modality we found significant positive correlations in the right dlPFC and more anterior area of the left dlPFC, and in the right NAc (however only the effect in the right dlPFC survived Bonferroni correction), and in the mechanical modality we found a significant positive correlation only in the more anterior region of the dlPFC (not surviving Bonferroni correction; see Supplementary Table 10 for full statistics). Thus, placebo-induced activation predicted weaker placebo effects in dlPFC, anterior PFC, and NAc.

## Discussion

Previous studies have provided mixed evidence on nociceptive modulation in placebo analgesia, with small, individual studies showing evidence for opioid release[42–46], spinal modulation[40,41], and effects in nociceptive brain regions, but meta- and mega-analyses revealing mostly small effects of placebo in nociceptive regions[51–54]. Here, in the

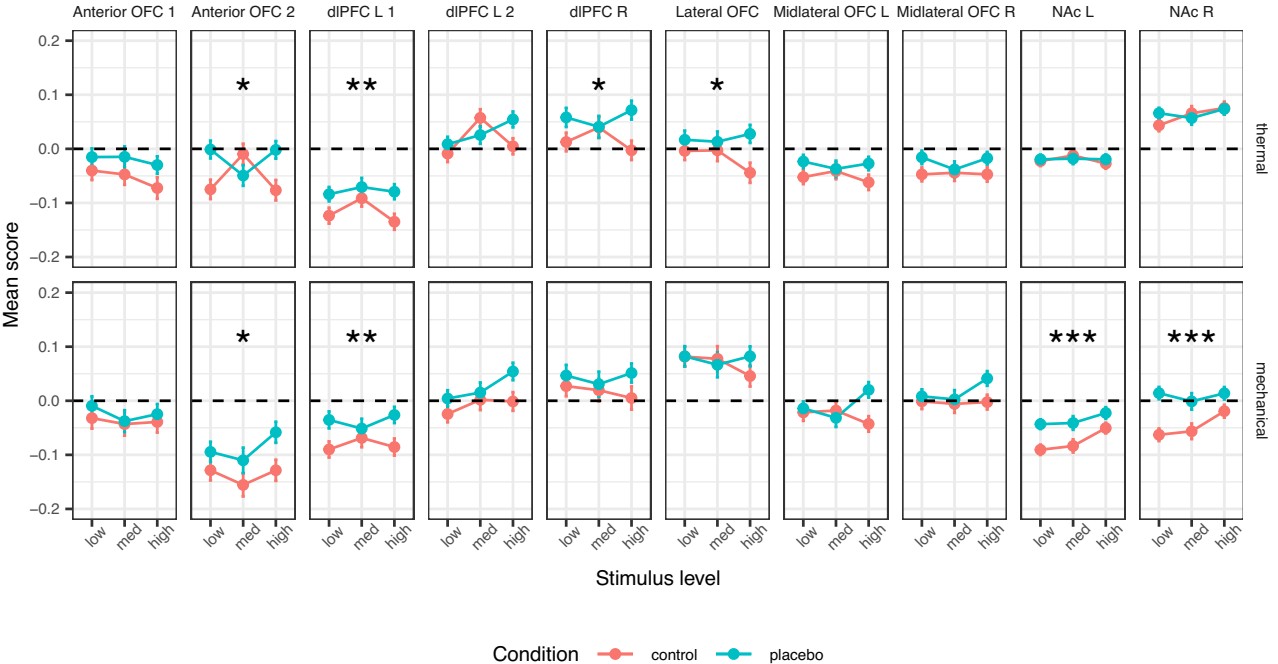

**Fig. 5 | Neural results: a priori higher-level pain processing ROIs.** The mean signal across all voxels of each a priori ROI across participants are presented, in the Control (red) and Placebo (blue) condition, for each combination of modality and stimulus level. Error bars represent within-participant standard error of the mean, based on Morey, 2008[107]. $N = 392$ participants. Asterisks represent significance of the placebo effect (mixed-effects model, uncorrected): $*p < 0.05$, $**p < 0.01$, $***p < 0.001$. For additional visualizations see Supplementary Fig. 7. L left, R right, dlPFC dorsolateral prefrontal cortex, OFC orbitofrontal cortex, NAc nucleus accumbens.

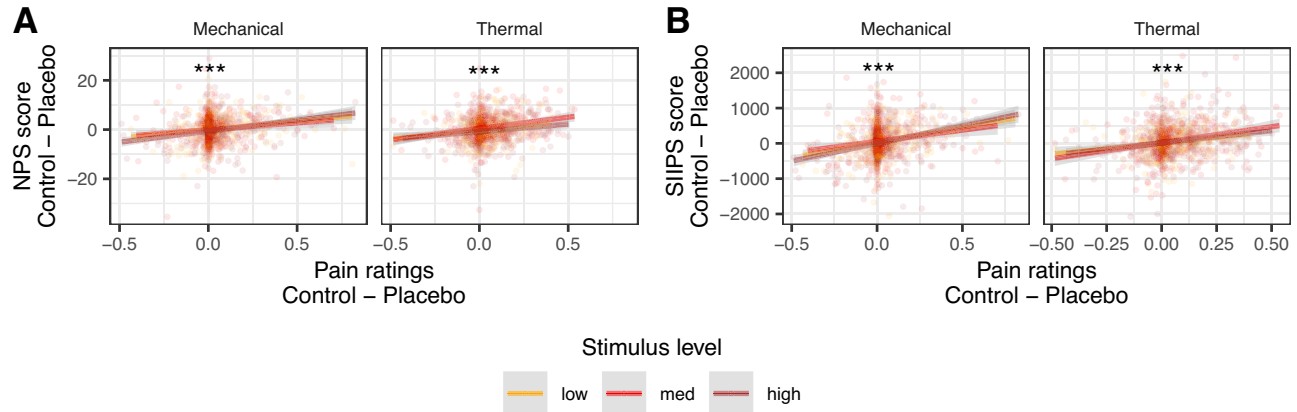

**Fig. 6 | Correlations between behavioral and neural placebo-induced reductions.** Placebo-induced neural reductions (Control - Placebo) in the (**A**) NPS or (**B**) SIIPS, as a function of the behavioral analgesia (pain ratings Control - Placebo) for each stimulus intensity level (color) and modality. Each dot is a participant. Lines represent linear smoother, with the 95% confidence interval shaded in gray. Asterisks represent significance of the correlation (mixed-effects model, uncorrected): $*p < 0.05$, $**p < 0.01$, $***p < 0.001$ (Thermal: $N = 367$ participants; Mechanical: $N = 368$ participants).

largest neuroimaging study of placebo effects to date, we investigated both conditioned placebo effects in thermal pain (the most common placebo paradigm) and transfer to unconditioned placebo effects in mechanical pain, a novel test of generalization. Placebo effects on reported pain were highly significant, and effect sizes were moderate ($d \sim= 0.5$), comparable to those found in previous studies, and were not significantly different between the conditioned heat pain modality and the mechanical pain transfer test.

In addition, placebo effects were modestly correlated across conditioned thermal and mechanical transfer modalities, but only correlated with pre-scan expectations in the thermal modality. These findings are important because how broadly placebo effects transfer across contexts and outcomes (e.g., pain responses) is a crucial and heavily debated issue[89–93,95,96,98–100]. On one hand, nociceptive pathways are organized by stimulus type, with different channels in the periphery[112,113] and somewhat divergent cortical representations[114,115] and pain sensitivity[116–119], and placebo effects are largely uncorrelated across pain types in some previous studies[89] (though these are likely underpowered given the effect sizes we report here). On the other hand, recent behavioral evidence suggests that placebo effects based on associative learning often do transfer to different modalities to produce benefits[92,98–100] and in some cases harms[95]. Our findings indicate robust behavioral placebo effects that are generalizable to new outcomes (i.e., a new pain modality). In addition, they show generalizability at the neural level for pain valuation and motivation-related brain processes, as we describe below.

Despite these substantial behavioral effects, fMRI analyses did not reveal significant reduction in the NPS–the most widely validated neuromarker of nociceptive pain to date–or nociception-related ROIs that were pre-registered based on previous studies. Bayes Factor analyses showed strong evidence in favor of null effects. Bayesian evidence for null effects was also found in individual subregions, precluding the possibility that the null effects resulted from a mix of positive and negative findings in different regions. In addition, in the novel test of transfer to mechanical pain, placebo treatment caused paradoxical activity increases in several of the regions most closely associated with pain processing, including aMCC, dpIns, and sensory thalamus (VPL/M; see below for discussion of these findings). The NPS has been shown to be sensitive to bottom-up stimulus intensity across multiple pain types[54,67], including sensitivity to both thermal and mechanical pain intensity in this study, and our sample size provided high power to detect small effects and provide strong Bayesian evidence in favor of the null. With $n = 392$ and $p < 0.05$, we have more than 80% power to detect "very small" effects of $d = 0.15$, and ~100% power to detect "small" effects of $d = 0.3$. The most recent meta-analysis revealed a significant placebo effect on the NPS, but this effect was very small ($g = 0.08$) and was found only in 3 out of 20 individual studies[54]. Moreover, of the nociceptive pain regions, a significant placebo effect was only found in a specific part of the insula, with a small effect size (and also the middle cingulate cortex when studies were modeled as a fixed rather than a random effect)[53]. Thus, together, these findings suggest that placebo analgesia is not driven by modulation of low-level nociceptive processes, at least for the average participant in experimental settings (for a discussion of individual differences see below). None or very small and condition-selective effects on early sensory construction make sense from a functional perspective. While early sensory modulation could be adaptive and energy efficient[22,24], it could also be dangerous–leading to false perceptions and hallucinations–and could impair learning by reducing prediction errors and error-driven learning in post-sensory processes, preventing the development of epistemically accurate internal models of the world.

Importantly, pain is more than nociception, and incorporates substantial affective and evaluative contributions[120]. We found evidence that placebo effects reduce activity in brain systems related to higher-level evaluative contributions to pain. The SIIPS neuromarker was trained to capture endogenous brain contributions to pain above and beyond stimulus intensity, reflecting these aspects. It is tempting to associate SIIPS with the affective aspects of pain, but strong correlations between pain affect (unpleasantness) and intensity measures were found in the current study and previous ones[121–123]; thus, we believe SIIPS is most closely associated with evaluative and value-construction aspects of pain. Here, the SIIPS response to thermal pain was reduced with placebo treatment, and this effect transferred to unconditioned mechanical pain. Effects in local regions were generally smaller, suggesting that reductions were distributed across brain systems, but particularly prominent reductions were found in the mechanical pain condition in the NAc, STG, and TP. The NAc has shown activation in previous meta-analyses of placebo[51,124], placebo-induced opioid release[45,46,125] and dopamine release[125], placebo-induced changes in reward prediction error signaling[126], and mediation of cognitive reappraisal effects on pain[127]. However, a recent meta-analysis showed evidence for NAc increases with pain but little evidence for placebo effects[53]. It may be that placebo-induced NAc increases are masked in many studies because placebo simultaneously directly activates and reduces pain-related inputs to NAc; future studies must disentangle the effects of pain of different types as well as placebo (or other 'top-down' effects) to test this further. STG and TP have seldom been discussed in the placebo literature, but they are extensively involved in the construction of emotional experiences and emotional memories[128].

In addition to effects on SIIPS, we found placebo-induced activity in dlPFC, anterior PFC, and OFC. These areas were suggested as key mediators of placebo effects in early studies[49] but effects were not consistent across studies in a recent meta-analysis[53]. Together, findings on the SIIPS, regions associated with high-level pain evaluation and construction of emotion, and prefrontal cortical increases support the conclusion that placebo analgesia is mostly driven by higher-level neural processing, including regions that are related to the construction of value and motivation.

The paradoxical placebo-induced increases in mechanical pain responses in some pain-related regions–aMCC, dpIns, and thalamus–warrant further discussion. Placebo-induced increases have not been reported in previous, smaller-scale studies to our knowledge, though these studies have not focused on transfer to unconditioned pain types. One possibility is that these increases reflect aversive prediction errors, an account broadly consistent with the predictive coding framework, with BOLD signals reflecting prediction errors from lower-level processing stages (i.e., pain-related signals that are not "canceled out" by predictions)[21,24]. Roy et al., 2014[129] and Geuter et al., 2017[71] provided some evidence for aversive prediction error coding in the insula and aMCC, and Roy et al. also found a paradoxical placebo-related increase in PAG. Combined with our findings of self-reported placebo analgesia, these findings suggest a dissociation between behavioral and neural correlates of pain. While pain reports integrate top-down predictions provided by the placebo context (i.e., assimilation effects), consistent with Bayesian accounts of placebo effects[24], nociceptive systems show the opposite effects (i.e., contrast effects), consistent with bottom-up signals that are not canceled out by predictions (because actual nociceptive input is higher than predicted under placebo treatment). This account is consistent with placebo effects on later-stage decision-making and experience construction rather than on early nociceptive signals.

This account is also consistent with a dual-process theory of pain modulation, in which safety cues that reduce the threat value of pain can have dual, opposing effects. On one hand, safety cues could allow competing motivational processes to gain priority and suppress pain, mediated in part by "motivation/decision" circuits like the NAc and medial PFC[130–132]. On the other hand, threat of pain induces preparatory processes that also suppress nociception (threat- and stress-induced analgesia), including endogenous opioid release[133–135]. Safety cues that reduce threat would block this effect. If threat analgesia has effects on lower levels of the nociceptive processing hierarchy than safety-related expectancies, placebo effects would increase nociceptive input (by reducing threat analgesia) while reducing pain valuation and motivation (in NAc/mPFC). Brain regions with a strong sensory component, like dpIns, sensory thalamus, and pain-selective portions of aMCC[136,137], would show paradoxical placebo-related increases such as those we observe here. Since these placebo-induced increases in nociceptive pain regions were only found in the transfer condition (i.e., in mechanical but not thermal pain trials), they may result from a transfer of the threat analgesia but no (or weaker) transfer of the safety-related expectancies.

Finally, this study provided a rare opportunity to examine individual differences in neural placebo effects in a sufficiently large sample to allow for stable correlation estimates and detection of small effects typical of between-person correlations[138,139]. We found significant correlations between behavioral analgesia and larger reductions in both the NPS and SIIPS, as well as some individual regions, including the aMCC, dpIns, PAG, and medial thalamus. This suggests that placebo treatment may cause reductions in the NPS and nociceptive brain regions in some individuals, as suggested by a previous person-level meta-analysis[53,54]. However, we cannot endorse this conclusion here because such correlations cannot provide strong evidence for causal effects. Here, as in previous literature, the strength of an individual's behavioral analgesic effect is conflated with random variation in

sensitivity on different skin sites and/or sensitization/habituation over time. For example, a participant whose placebo skin site is less sensitive than the control skin site is expected to report less pain for placebo compared to control irrespective of the placebo effect induced by the placebo manipulation, and also to have a lower NPS response for placebo compared to control. Because skin site sensitivity across participants is random, and skin site order was counterbalanced across participants, this random variation should not influence the main effect of placebo in the present study. It could, on the other hand, induce correlations between control minus placebo differences in pain ratings and neural responses, as we have found. This issue is common to virtually all clinical trials of treatment effects. Future studies could productively examine brain-behavioral correlations by selecting high and low placebo responders based on independent criteria (e.g., a separate session using different skin sites). Here, pre-scan expectation ratings could serve this goal, but expectancy ratings were not correlated with neural placebo-control differences, and were only associated with behavioral analgesia in thermal pain.

Individual difference correlations across thermal and mechanical pain do not suffer from this problem (because different skin sites were used for each modality). However, these correlations were significant behaviorally but non-significant for both NPS and SIIPS, suggesting low trait-like stability of neural placebo responses. This is important because the ability to predict who will be a placebo responder is a crucial and perennial issue in medicine–with many attempts to predict and control for placebo effects in drug and device trials[140,141]–and such predictions hinge on the ability to identify stable traits across outcomes and time. Whereas some studies have found correlations between placebo effects and trait-like measures including genetics[142–144], brain structure, and personality[145,146], most of these studies have been small and underpowered for the data types and expected effect sizes[138,147] (but see ref. [148]), and a recent meta-analysis of personality shows inconsistent effects across studies[91]. One of our initial hypotheses was that while self-reported placebo effects and personality may be unstable across contexts (e.g. ref. [90]), neural responses would be more stable at the individual differences level and show stronger correlations across pain types, but this did not appear to be the case here. In addition, while previous studies have found sex differences in placebo effects (though these differences were inconsistent across studies and interacted with other factors)[73–75], here we did not find significant sex differences in placebo analgesia (see Supplementary Information).

There are several additional limitations to the current study that should be taken into account. First, the overall pain level across participants was relatively low based on the subjective pain ratings, with the vast majority of ratings indicating less than a moderate amount of pain, particularly in the mechanical modality. These may have been the result of the specific stimulus intensities chosen (which had to be painful to participants while moderate on average to be tolerable to a broad population without participant dropout and sampling bias), or of using the finger as the body site of stimulation, since glabrous skin lacks type II aδ fibers[149]. Nevertheless, we observed strong placebo effects on pain ratings and placebo effects on SIIPS and several individual regions, as well as substantial variability across participants that allowed us to test for individual differences. Second, the use of an imaging protocol with a multiband acceleration factor of 8 may have enhanced signal dropout in some areas, such as ventral prefrontal areas and NAc[150], though we did find effects in some of these regions. Third, we used the canonical hemodynamic response function (cHRF), as was pre-registered and done in most previous studies. However, early studies[43,49] have distinguished between early and late responses without assuming cHRF, and different HRF models may yield different results. Fourth, different placebo-induction protocols or analysis pipelines may lead to different findings[151], for example with regard to the effect of placebo on nociceptive processes. Here, the placebo

induction combined several components in order to maximize placebo effects[152] (e.g., suggestion, conceptual conditioning[84,153] and classical conditioning)[24,154], which yielded strong behavioral effects, in addition to an unconditioned transfer condition (which produced similarly large behavioral and neural effects). More studies are needed in order to systematically compare different components of the paradigm and test whether different brain systems are involved in the induced placebo effect (see for example refs. [152,155]). In addition, the analysis pipeline used is a common one and many of its components were pre-registered. Fifth, the majority of participants in the current study were white adults. Further studies will be needed to test whether its results generalize to samples with other demographic characteristics[76] or different contexts. Sixth, like any study manipulating participants' expectations, demand characteristics may drive participants to report less pain in the placebo condition. The size of the placebo effect on pain ratings was comparable to previous studies (e.g., $d = 0.53$ here and $g = 0.66$ in Zunnhammer et al.[54]), suggesting that there is not an extra demand characteristic effect in the present study. Furthermore, we revealed significant placebo effects on the SIIPS, and pre-scan expectations were not correlated with this effect across participants, suggesting that these neural effects do not represent effects of demand characteristics. Future studies are needed to examine which types of evaluative contributions to pain are reflected in the SIIPS, and whether they are trivial ("biases") or consequential (long-term changes in pain decision-making and behavior).

Though there are strengths of the neuromarker approach in power and specificity to pain, there are also limitations, which in part motivated additional analyses on individual regions. First, neuromarkers like the NPS and SIIPS do not explicitly test signals in any one brain region and their neuroanatomical meaning is thus more difficult to interpret. Testing subregions provides additional clues about which regions are affected by placebo individually. Second, these neuromarkers do not capture pain equally well in all participants, and there are variations across individuals in the brain bases of self-report[156]. Unimodal somatosensory regions tend to be the most consistent across participants, whereas transmodal regions like the vmPFC and lateral PFC have more variable pain representations across participants. Third, it is also possible that placebo could change the spatial topography of activation rather than reducing the magnitude of activation in defined neuromarkers. Fourth, these neuromarkers are designed to capture variation in pain reports in healthy participants, and are not designed to capture all contributions to pain in all situations. Finally, the specificity to pain of the SIIPS in particular is less well validated, and it may be more affected by other affective and cognitive processes that are not directly linked with pain.

Overall, addressing the critical question of the neural level at which placebo manipulations operate, our findings strongly indicate that a combination of conditioning and suggestion leads to meaningful reductions in reported pain that are manifested in neural changes in higher-level brain regions related to pain, decision-making, and motivation, but not lower-level nociceptive pain brain regions. These effects are not limited to conditioned effects, but transfer to outcomes that utilize at least partially distinct neural circuitry. More broadly, our findings point at the importance and promise of new treatments that are based on beliefs and expectations, in pain and beyond, operating on higher-level neural processes that could yield meaningful clinical improvements[157,158]. They also indicate that neuromarkers can help separate treatment influences on nociception from influences on symptom construction and decision-making, and that the NPS is a promising placebo-insensitive brain target for treatment development. Such new treatments are critically needed now even more than ever, as pain remains an essential but poorly understood motivating force and chronic pain remains undertreated. Continued studies of placebo effects are needed to understand how the brain constructs

pain experience and how those experiences drive our long-term experience and behavior.

## Methods

### Participants

Participants were recruited by telephone from the Colorado Community Twin Sample, which is derived from the Colorado Twin Registry, a population-based registry which has been run by the Institute of Behavioral Genetics (IBG) at the University of Colorado since 1984[159]. The study includes a larger sample of participants, but this study is based on a pre-registered dataset of 397 participants who completed their participation and whose data were preprocessed by 1 September 2022, when we pre-registered the study (pre-registration link: https://osf.io/unh7f). Participants were excluded from participation in the study if they did not pass MRI screening (e.g., add metals in their body) or had a history of liver disease/damage, allergies to local analgesics, or were breastfeeding.

Some participants have been excluded prior to the pre-registration due to a protocol change early in the study ($n = 11$), data loss/corruption ($n = 11$), errors related to our thermal stimulator ($n = 1$) or inconsistencies in the placebo induction/treatment protocol ($n = 3$). Two participants who were included in the pre-registered sample were excluded from analyses because of corrupted behavioral data. The final dataset included 395 participants, of which 142 are monozygotic twins, 160 are dizygotic twins, and 93 are individuals without siblings in the dataset. Participants' age ranged from 30 to 43 years ($M = 35.43$, $SD = 2.60$), and the sample included 163 men and 232 women, and was predominantly white (346 White, 15 Hispanic/Latino, 8 Asian, 5 mixed, 3 Native American, 1 Black/African American, 1 Pacific Islander, 16 unknown). All participants provided their informed consent at the beginning of the experiment and received monetary compensation for their participation. The experiment was approved by the institutional review board of the University of Colorado Boulder.

Individual stimulus trials were excluded if the response time of the pain intensity rating was above 5.01 or below 0.02 seconds. These indicate either that the participant did not respond within the time allotted (five seconds) or responded too quickly to represent deliberate ratings. Entire participants were dropped if after dropping trials or scans due to above exclusion criteria, they lacked complete sets of conditions (placebo-thermal, placebo-mechanical, control-thermal, control-mechanical) for more than one stimulus level. Overall, 392 participants were included in the neuroimaging analyses, and 374 participants were included in the behavioral analyses.

### Procedures

First, participants were presented with an overview of the experiment components, completed a short survey about their health and mood, completed an anti-saccade task, provided saliva samples, and practiced the rating scale, with two ratings per stimulus: intensity rating and unpleasantness rating. To distinguish between the two ratings, participants were given an example: when listening to a song on the radio, as the volume increases, the intensity is the loudness of the music, while the (un)pleasantness is how much they like the song. We used a Labeled Magnitude Scale (LMS), consisting of quasi-logarithmically spaced perceptual verbal labels (0.014 = barely detectable, 0.061 = weak, 0.172 = moderate, 0.354 = strong, 0.533 = very strong[105,106]), which provides ratio properties[160] and avoids ceiling effects on pain reports that are common with some narrow-range rating scales. Participants were presented with the intermediate scale labels during training, but they were removed when obtaining actual pain ratings during the experimental tasks, to minimize clustering around the labels[161].

After the scale's training, participants completed a familiarization task, to ensure their tolerance to the painful thermal stimuli. Stimuli in the familiarization task ranged between 45.5 and 48.5 °C, with a duration of 10 seconds each. Participants rated each stimulus verbally based on the same LMS scale. Participants were instructed that any sensation they would describe as pain should get an intensity rating above 0, and that the most intense pain they would normally tolerate should be rated between 0.5-0.6. Participants who were unable to tolerate the stimuli were excluded from further participation. The thermal stimuli were delivered with a 16 ×16 mm surface thermode (PATHWAY ATS; Medoc, Inc, Israel). The skin site used for the familiarization task was the right index finger.

Following the familiarization task, the placebo manipulation took place. Participants were given two identical creams with different instructions. One cream was introduced as "Prodicaine, an effective pain-relieving drug" (the placebo cream), and the second cream was introduced as "a control cream with no effects" (the control cream). These two creams were applied to two different fingers of the left hand of the participant (cream allocation to fingers was counterbalanced across participants, and the cream was applied to the entire inner part of each finger). Participants watched a video describing the Prodicaine application procedure and a short testimonial from an (allegedly) pilot participant. They also received forms disclosing potential side effects of the Prodicaine, with realistic-looking drug company logos, from research assistants wearing professional attire and lab coats, in a room with medical equipment and related contextual cues.

To strengthen expectations of pain relief from the placebo treatment cream, participants were subjected to two conditioning paradigms[152]. Importantly, both conditioning paradigms were based on thermal, not mechanical, stimuli. First a "symbolic" conditioning paradigm was administered. This paradigm was similar to previous studies[84,153]. An inert thermode was placed on the control skin site and the participants completed a trial sequence mimicking a thermal stimulation sequence, except instead of thermal stimuli they were shown ratings they were told come from prior participants. These ratings were systematically high for 16 stimuli. The thermode was then moved to the prodicaine treated site and the procedure was repeated for 32 stimuli. This time the ratings were systematically low. Finally the thermode was moved back to the control site and 16 additional ratings were once again systematically high. The thermode was placed on the proximal phalanges. Second, a classical condition paradigm was administered. This paradigm was identical to the symbolic conditioning paradigm, except instead of being shown ratings of other participants, participants were subjected to noxious thermal stimulation of the proximal phalanges. They were told that stimuli were all of the same intensity, but the intensity was surreptitiously lowered by 3.5 °C when stimulating the Prodicaine treated skin site compared to the control treated skin site (44 and 44.5 °C for the placebo cream and 47.5 and 48 °C for the control cream). Following each stimulus, participants rated the intensity (from "no pain" to "most pain imaginable") and unpleasantness (from "not at all" to "worst pain imaginable") of the stimulus. After the conditioning task, participants rated their expectations regarding the Prodicaine efficacy in the next task, on a linear scale between 0 [not at all] to 100 [most effective].

Following the placebo manipulation, the test task was conducted in the MRI scanner. The creams were reapplied to four fingers (index, middle, ring, little finger), two of which to be used for thermal and two for mechanical (unconditioned) stimuli, with one finger for the placebo cream and one for the control cream in each modality (fingers allocation was counterbalanced across participants). Mechanical pain stimuli were administered using an in-house pressure pain device, an MRI-safe device with dynamic pressure delivery controlled by LabView (National Instruments). The test task included 32 stimuli divided into four runs, with four thermal and four mechanical stimuli in a random order per run. Three stimulus intensities were used for each modality: low, medium, and high (thermal: 46.5, 47 and 47.5 °C; mechanical: levels 3,4,5, corresponding to about 6, 7 and 8 kg/cm², respectively). Half of the stimuli from each modality were delivered to the finger with

the control cream, and half to the finger with the placebo cream, such that the first and last run included stimulation of the control cream skin site, and the second and third included stimulation of the placebo cream skin site (a control-placebo-placebo-control design). Each stimulus lasted 10 seconds, and was delivered to the distal phalanges of the left hand. Before each stimulation, an anticipatory cue was presented for 0.5 second followed by a 1 or 3 s randomly-jittered delay. Three seconds after each stimulus, participants rated its intensity and unpleasantness (5 s each, in a random order) with a tracking ball.

## Imaging data acquisition

Imaging data were acquired using a 3 T Siemens Prisma MRI scanner with a 32-channels head coil, at the University of Colorado at Boulder. First, a T1-weighted structural scan was acquired using a magnetization prepared rapid gradient echo (MPRAGE) pulse sequence with parallel imaging factor (iPAT) of 3, TR = 2000 ms, TE = 2.11 ms, flip angle = 8°, FOV = 256 mm, resolution = $0.8 \times 0.8 \times 0.8$ mm. Then, we acquired fieldmaps for each participant, one with a posterior-anterior (PA) and one with an anterior-posterior (AP) direction, with the following imaging parameters: TR = 7220 ms, TE = 73 ms, flip angle = 90°, FOV = 220 mm, and in plane resolution of $2.7 \times 2.7 \times 2.7$ mm. Resting-state fMRI data was acquired using T2*-weighted echo-planar imaging (EPI) sequence with multiband acceleration factor of 8, TR = 460 ms, TE = 27.20 ms, flip angle = 44°, FOV = 220 mm, resolution of $2.7 \times 2.7 \times 2.7$ mm, 56 slices, and 816 acquired volumes. Then, participants completed four runs of the pain test task while scanned with a similar fMRI sequence, obtaining 550 volumes for each of the four task fMRI runs. They then completed a perspective-taking fMRI task (similar protocol, 1322 volumes; this task is outside the scope of the present paper). Finally, two diffusion-weighted (dMRI) scans were acquired, one with a PA and one with an AP acquisition direction. The dMRI PA scan included a multiband acceleration factor of 3, 47 directions, TR = 4000 ms, TE = 77 ms, FOV = 224 mm, flip angle = 84°, with a b-value of 2400 s/mm$^2$. The AP dMRI scan was similar, except for an inverse acquisition direction and 44 diffusion directions acquired.

## Imaging data preprocessing

Structural and functional data were preprocessed using fMRIPrep version 20.2.3[162] (RRID:SCR_016216), which is based on *Nipype* 1.6.1 (RRID:SCR_002502; ref. [163,164]).

**Anatomical data preprocessing.** The T1-weighted (T1w) image was corrected for intensity non-uniformity (INU) with N4BiasFieldCorrection[165], distributed with ANTs 2.3.3 (RRID:SCR_004757; ref. [166]), and used as T1w-reference throughout the workflow. The T1w-reference was then skull-stripped with a *Nipype* implementation of the antsBrainExtraction.sh workflow (from ANTs), using OASIS30ANTs as target template. Brain tissue segmentation of cerebrospinal fluid (CSF), white-matter (WM), and gray-matter (GM) was performed on the brain-extracted T1w using fast (FSL 5.0.9, RRID:SCR_002823; ref. [167]). Brain surfaces were reconstructed using recon-all (FreeSurfer 6.0.1, RRID:SCR_001847; ref. [168]), and the brain mask estimated previously was refined with a custom variation of the method to reconcile ANTs-derived and FreeSurfer-derived segmentations of the cortical gray-matter of Mindboggle (RRID:SCR_002438; ref. [169]). Volume-based spatial normalization to two standard spaces (MNI152NLin2009cAsym, MNI152NLin6Asym) was performed through nonlinear registration with antsRegistration (ANTs 2.3.3), using brain-extracted versions of both T1w reference and the T1w template. The following templates were selected for spatial normalization: *ICBM 152 Nonlinear Asymmetrical template version 2009c* (RRID:SCR_008796; TemplateFlow ID: MNI152NLin2009cAsym; ref. [170]], *FSL's MNI ICBM 152 non-linear 6th Generation Asymmetric Average Brain Stereotaxic Registration Model* (RRID:SCR_002823; TemplateFlow ID: MNI152NLin6Asym[171]).

**Functional data preprocessing.** The single-band reference (SBRef) was used as a reference volume along with its skull-stripped version. A B0-nonuniformity map (or *fieldmap*) was estimated based on two EPI references with opposing phase-encoding directions, with 3dQwarp (AFNI 20160207; ref. [172]). Based on the estimated susceptibility distortion, a corrected EPI reference was calculated for a more accurate co-registration with the anatomical reference. The BOLD reference was then co-registered to the T1w reference using bbregister (FreeSurfer) which implements boundary-based registration[173]. Co-registration was configured with six degrees of freedom. Head-motion parameters with respect to the BOLD reference (transformation matrices, and six corresponding rotation and translation parameters) are estimated before any spatiotemporal filtering using mcflirt (FSL 5.0.9[174]). First, a reference volume and its skull-stripped version were generated using a custom methodology of *fMRIPrep*. The BOLD time-series were resampled onto the following surfaces (FreeSurfer reconstruction nomenclature): *fsnative, fsaverage6, fsaverage*. The BOLD time-series were resampled onto their original, native space by applying a single, composite transform to correct for head-motion and susceptibility distortions. These resampled BOLD time-series will be referred to as *preprocessed BOLD in original space*, or just *preprocessed BOLD*. The BOLD time-series were resampled into standard space, generating a *preprocessed BOLD run in MNI152NLin2009cAsym space*. First, a reference volume and its skull-stripped version were generated using a custom methodology of *fMRIPrep*. All resamplings can be performed with *a single interpolation step* by composing all the pertinent transformations (i.e. head-motion transform matrices, susceptibility distortion correction when available, and co-registrations to anatomical and output spaces). Gridded (volumetric) resamplings were performed using antsApplyTransforms (ANTs), configured with Lanczos interpolation to minimize the smoothing effects of other kernels[175]. Non-gridded (surface) resamplings were performed using mri_vol2surf (FreeSurfer). Following the preprocessing with fMRIPrep, data were smoothed with a Gaussian kernel of 6 mm.

## Data analysis

The first level model included the following regressors: four regressors for the cue period (one for each combination of modality [thermal / mechanical] and condition [placebo / control]), 12 regressors for the pain-evoked period, and a regressor for the rating period. We modeled 12 experimental conditions for the pain-evoked period in a $3 \times 2 \times 2$ factorial design–including Stimulus level (3 levels), Modality (2 levels, thermal and mechanical), and Placebo treatment (2 levels, placebo and control)–using a separate regressor for each condition. Indicator vectors ([0,1]) indicating the presence or absence of each condition were convolved with a canonical hemodynamic response function implemented by spm12. Nuisance regressors included 24 motion regressors (six motion parameters–translation and rotation in three directions–together with their derivatives, quadratics and derivatives of the quadratics) and a mean CSF signal regressor (estimated by fMRIPrep during preprocessing). We also implemented spike censoring, with spikes identified by our in-house spike detection algorithm implemented by CanlabCore/diagnostics/scn_session_spike_id.m (available at github.com/canlab/CanlabCore). The first eight frames of each run were censored, and a 180 Hz high pass temporal filter was implemented using cosine basis functions. This procedure resulted in 12 separate parameter estimate maps for each stimulus condition (e.g., high intensity thermal stimuli delivered to the placebo skin site), for each participant.

Statistical analysis was performed using Matlab 2020a and 2021b, SPM12, and CANlab neuroimaging analysis tools (shared via Github at https://canlab.github.io/; the version used was from the beginning of August 2022, with the last commit of Canlab Core being b6db85e, SHA b6db85e2577d967d90c3bbe508c3c6acff37e268). Activity in each a priori region of interest or neuromarker for each contrast (each

condition of each participant) image was computed as a continuous score. For a priori regions of interest, the score was based on the averaged univariate GLM-derived BOLD contrast estimates across voxels within the region of interest. For neuromarkers and their sub-regions, the score was based on the dot product of the univariate map with the neuromarker weight map.

The different scores, as well as the behavioral pain ratings, were tested at the group level using a mixed effects model. In the case of datasets with twins, treating individuals as independent observations leads to inflated estimates of degrees of freedom and consequently overly liberal statistical inference. To account for this, we introduced familial random effects (i.e., of twin dyad) into the model. Thus, fixed effects can be interpreted as the mean effect taken over families rather than individuals. Twin studies additionally introduce heteroscedastic sources of variance since, within families, dizygotic twins are expected to be more variable than monozygotic twins. Naive maximum likelihood methods would produce fixed effect parameter estimates that are biased towards the dizygotic twins in the sample. To prevent that, we instead model each category of twin using separate random effects covariance parameters[176]. Our implementation used Matlab R2020a's fitlme function to estimate effects of the placebo condition and the stimulus intensity. Formally, using Wilkinson notation[177]:

score ~ 1 + stimulus_level + placebo_condition + (1 + stimulus_level + placebo_condition | family_ID) + (intercept_monozygotic + stimulus_level_monozygotic + placebo_condition_monozygotic −1 | participant_ID) + (intercept_dizygotic + stimulus_level_dizygotic + placebo_condition_dizygotic − 1 | participant_ID).

The stimulus_intensity was coded as −0.5, 0, 0.5 for low, medium, high, respectively, and the placebo_condition was coded as −0.5 for control and 0.5 for placebo. All regressors with the suffix "_dizygotic" were 0 for monozygotic twins, and all regressors with the suffix "_monozygotic" were 0 for dizygotic twins. The random effects covariance matrix was also constrained to be block diagonal by zygosity, such that the covariances of parameters with the suffix "_monozygotic" and with the suffix "_dizygotic" are zero. This generalizes a heteroskedastic twin error model[176] to within participant repeated measures designs.

Each model was tested with the stimulus level x placebo condition interaction term as a fixed effect, and then without the interaction term if the interaction was not significant. In such a case, to increase interpretability, the mean effects reported in the main text were based on the model without the interaction. Note that the interaction was not significant for all tested neuromarkers and regions of interest. Data from each pain modality were tested separately. Subjective pain ratings and brain scores within each modality were z-scored across participants (maintaining within- and between-subject effects). Satterthwaite's method[178] was used to estimate degrees of freedom for the mixed effects models. As pre-registered, participants without full data for at least two stimulus levels were excluded from the analysis. This led to the exclusion of 22 trials (0.47% of total number of trials), from three participants. Overall, 392 participants were included in the models for the effects of placebo and stimulus intensity. When the effect size is presented in the text as d, it is based on the ratio between the mean / sd across participants, without accounting for the familial structure.

To test the correlation between the behavioral and neural placebo-induced reductions, we computed the averaged pain rating or brain score for each participant in each stimulus modality and intensity level separately for the placebo and the control condition. We then computed the difference between these scores (control minus placebo) within each combination of modality and stimulus level for each participant, and z-scored the values across participants within each modality. We also computed the rank order of differences (not z-scored) within each combination of stimulus level and modality.

These values were inputted to the following mixed-effects model, computed again with fitlme in Matlab 2020a, separately for the thermal and mechanical pain modalities:

brain_score_difference ~ 1 + stimulus_level + behavioral_analgesia + (1 + stimulus_level + behavioral_analgesia | family) + (stimulus_level_monozygotic + behavioral_analgesia_monozygotic + intercept_monozygotic −1 | participant_ID) + (stimulus_level_dizygotic + behavioral_analgesia_dizygotic + intercept_dizygotic −1 | participant_ID).

To directly compare the placebo-induced reduction between the NPS and SIIPS, we first z-scored the scores of each neuromarker across conditions (placebo/control and stimulus intensity) and participants, such that neuromarkers could be represented on the same standardized scale but the within and between-subject effects within each neuromarker were not affected. Then, we computed the control - placebo difference for each neuromarker for each combination of participant, modality and stimulus level, and compared NPS vs. SIIPS with the following mixed-effects model:

brain_score_difference ~ 1 + neuromarker + stimulus_level + (1 + neuromarker + stimulus_level | family_ID) + (neuromarker_monozygotic + stimulus_level_monozygotic + intercept_monozygotic −1 | participant_ID) + (neuromarker_dizygotic + stimulus_level_dizygotic + intercept_dizygotic −1 | participant_ID).

All statistical tests were two-sided. Note that we report results based on $p < 0.05$, without correcting for multiple comparisons. This is because in the current paper there is a need to balance between type 1 and type 2 errors, as we are testing multiple a priori (and mostly pre-registered) brain signatures and regions of interest based on previous literature with a new, substantially larger sample. To be slightly more conservative, we further note when a specific result does not survive Bonferroni correction for multiple comparisons within a specific set of regions (i.e., within the nociceptive regions, within SIIPS subregions, or within the set of higher-level regions).

Bayes Factor analysis was performed using R version 4.2.2., with the BayesFactor package version 0.9.12-2. Bayes factors were computed as the ratio of evidence in favor of the alternative hypothesis, which is the full model including main fixed and random effects (without random slopes), and the null hypothesis, which is the same model without the fixed effect of interest (e.g., the placebo effect[108]). For the prior distribution, we used a wide Cauchy prior distribution (rscale value of 1). We further tested the robustness of the Bayes Factor results to different scaling factors of the prior distribution ($\sqrt{2}/2$, 1, and $\sqrt{2}$) and to the inclusion of an interaction term in the alternative model.

In addition, we tested the correlations between pre-scan expectations of Prodicaine efficacy and placebo effects on the behavioral pain ratings, NPS score, and SIIPS score. The correlations were tested with a mixed-effects model similar to the one described above for the correlations between behavioral analgesia and placebo-induced reductions in brain responses, with the participant-level expectations as a predictor instead of the behavioral analgesia. We again tested these correlations separately for the thermal and mechanical modalities, and the expectations were z scored within each modality across participants. Correlations between placebo effects in the thermal and mechanical modalities were computed with a mixed-effects model predicting placebo-induced analgesia (or placebo-induced reductions in NPS or SIIPS score) in the mechanical condition based on the placebo-induced analgesia in the thermal condition (with the full random effects structure at the family and participant level as in the other models described above).

## Pre-registration

Pre-registration was performed following preprocessing of the imaging data, but prior to performing the data analysis that informs the hypothesized outcomes. The behavioral analyses were not pre-registered, as the pre-registration focused on the neural predictors and

correlates of placebo analgesia, assuming a behavioral effect of placebo analgesia as has been repeatedly shown in numerous previous studies. In addition, the pre-registration did not include the NPS tests. This is because the NPS analysis was already run on some of the participants prior to the submission of the pre-registration, and therefore could not have been considered a non-tested hypothesis like the other hypotheses. However, the test of the placebo effect on the NPS score was run on the full pre-registered sample only after the pre-registration was submitted. The Bayes Factor analysis was also not pre-registered, but was used to quantify evidence in favor of the null vs. alternative hypothesis, particularly when null hypothesis significance testing (NHST) results were not significant and testing potential evidence in favor of the null was important. Finally, the pre-registration included a long list of additional hypotheses regarding a priori neural signatures and brain patterns, as well as a comparison of early vs. late activity during pain period, and tests of heritability of effects. These analyses are beyond the scope of the current paper, and will be executed and published in future papers.

### Reporting summary
Further information on research design is available in the Nature Portfolio Reporting Summary linked to this article.

## Data availability
All behavioral and neuroimaging data included in this paper are shared on OpenNeuro: ds004746; doi:10.18112/openneuro.ds004746.v1.0.0.

## Code availability
Analysis codes are publicly shared on Github: https://github.com/rotemb9/paingen-placebo-fmri-paper (release 2.0.0). CANlab neuroimaging analysis tools that were used as part of the analysis are available at https://canlab.github.io/ (see methods for the version used).

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

## Acknowledgements

The study was funded by a National Institutes of Health (NIH) grant (DA046064, PIs N.P.F. and T.D.W). The funders had no role in study design, data collection and analysis, decision to publish or preparation of the manuscript. The authors thank Matthew Keller for helpful discussions. Thanks to Gordon Matthewson, Dan Kusko, Dan Ryan, Patricia Townsend, Clayton Schneider, Mickela Heilicher, Suebin Song, Abi Adams, Jia Moore, Alexa Gonzalez, and Corinne Gunn for helping with data collection. Rotem Botvinik-Nezer thanks the Golda Meir Fellowship.

## Author contributions

Based on Contributor Rules Taxonomy (CRediT): Conceptualization: all authors. Data curation: R.B-N., B.P., and M.C. Formal analysis: R.B-N. and B.P. Methodology: All authors. Funding acquisition: N.P.F. and T.D.W. Investigation: B.P. and M.C. Project administration: M.C. Software: R.B-N., B.P., and T.D.W. Supervision: N.P.F. and T.D.W. Visualization: R.B-N., B.P., and T.D.W. Validation: R.B-N. and B.P. Writing – original draft: R.B-N., B.P., and T.D.W. Writing – review & editing: all authors.

## Competing interests

The authors declare no competing interests.
