## [Peer Review File · Nature Communications]

Placebo treatment affects brain systems related to affective and cognitive processes, but not nociceptive painEditorial Note: This manuscript has been previously reviewed at another journal. This document only contains reviewer comments and rebuttal letters for versions considered at *Nature Communications*. Mentions of prior referee reports have been redacted.

REVIEWER COMMENTS

Reviewer #2 (Remarks to the Author):

My comment has been appropriately addressed. Excellent paper.

Reviewer #3 (Remarks to the Author):

The study by Botvinik-Nezer et al. is an investigation to deeply investigate mechanisms of placebo using activation signatures that hold the potential to be more sensitive to activation than traditional massive univariate approaches. Furthermore this investigation seeks to better test the transfer of placebo conditioning from one modality (heat) to mechanical stimulation. The tremendous efforts of the authors to collect a huge ($n=392$) fMRI dataset to address this important question are certainly appreciated. However, numerous major issues with pain ratings, stimulus delivery, and the conditioning paradigm raise serious questions about their central conclusion that higher order cognitive/affective brain regions but not nociceptive processing regions mediate the placebo response.

The central conclusion is based on the observation that the NPS was not significantly related to the placebo response, while the SIIPS was. However, there is no direct statistical comparison of the NPS vs SIIPS signals to support this conclusion. Was the SIIPS more related to the placebo response than the NPS? Moreover, the incorporation of individual differences into the NPS revealed that reductions in perceived pain intensity were, in fact, related to reductions in the NPS and nociceptive ROI's, bringing the findings more into line with existing studies.

In studies of human pain mechanisms, obtaining accurate and precise reports of pain is critical. There are a number of well-described and well validated scales in the literature. Many are relatively continuous and have interval or ratio scale properties essential for use as a covariate of interest to interpret brain activity. In this study, the scales for rating pain are barely described and it appears that two separate scales are used. One is used in the "calibration" phase and appears to be a verbal numerical scale. The other used in the main part of the study is some form of trackball controlled scale that appears to have verbal descriptors positioned at certain numbers and/or positions on the scale. The Bartoshuk reference is a review article that describes multiple types of scales and does not provide a description or validation of the scale that appears to be used during the fMRI portion of the study. Moreover, the instructions to distinguish intensity and unpleasantness are neither described nor validated.

The pain ratings of both the thermal and mechanical stimuli are quite low, with ratings of only .191/1 for heat and 0.13/1 for mechanical. This raises major questions about whether sufficient brain activation by these apparently weak stimuli was to detect a placebo reduction. This becomes particularly important when the main conclusion is that the nociceptive network was not modulated but the more cognitive/affective network was modulated by placebo. Could this just be a floor effect since the ratings were so low to begin with? This appears to contradict a relatively large body of placebo literature – including the very impressive meta-analysis by Zunhammer et al. (and some of the present authors) showing involvement of the medial and lateral thalamus, posterior insula, ACC, and other regions that have direct spino-thalamic input and are associated with nociceptive processing. Also, the ratings in the text are lower than those in figure 2 although they are both supposedly means of the same data.

The low ratings may arise from two factors. First, glabrous skin can be a great deal less sensitive

to noxious stimuli than hairy skin – likely due to the absence of some populations of A-delta nociceptors. Thus, the noxious stimuli may have simply not been sufficiently intense to elicit much of a perceptual response and concomitant activation of nociceptive processing mechanisms. Second, the measurement properties of the pain scale are unknown based on the absence of any citations regarding the validation of this scale (or scales since two seemed to have been used). Is it possible that the positioning of the labels both drove ratings down and caused them to cluster in specific positions? The SEMs of the ratings are miniscule.

Another major concern of the experimental design is that there were two forms of placebo conditioning used – a symbolic and a classical conditioning. The symbolic conditioning paradigm is of significant concern given the results. This paradigm involved probe placement and then presentation of fake example ratings for the control vs placebo condition. This raises concerns that demand characteristics of this procedure would lead to significant biases of ratings to conform to those provided by the fake participants rather than to reflect what participants actually felt. Thus, was the differential activation in the cognitive/affective areas related to rating biases and/or the formulation of ratings rather than actual modulation of the experience of pain?

Moreover, the use of two placebo paradigms in series seems likely to further complicate interpretation of the results and would likely be largely unnecessary given the demonstrated strength of the classical conditioning paradigm. Would the symbolic conditioning paradigm engage different mechanisms than the classical conditioning paradigm? Would some participants use one vs the other, or even both together?

Another concern regarding the lack of modulation of nociceptive processing regions arises from the relatively minimal number of trials involving noxious stimulation. There were 6 stimuli (low, med, high X thermal vs mechanical) and there were 8 stimuli per run, with two control runs and two placebo runs. Thus, most stimuli were only repeated twice during each control and each placebo condition. Also, other temperatures are mentioned during the conditioning procedure. Were these also used in the fMRI portion? Detecting change in this very complicated design (3x2x2) with perceptually weak stimuli and minimal repeats of individual stimuli seems challenging.

The “calibration” procedure is hopefully misnamed. For most researchers in the field, “calibration” would mean that stimuli are individually adjusted for each participant based on their ratings. This process introduces huge complexity in since different stimuli are used in different individuals, different populations of primary afferents are likely activated, with unknown impact on brain activity or the ability of pain to be modulated. In contrast, “training” or “familiarization” would imply that participants were exposed to the stimuli to see if they could tolerate them and to gain experience providing pain ratings. Unfortunately, it seems that different pain scales were used during familiarization and fMRI, so there was no benefit of training with scale usage. Also the site of “calibration” was not defined.

Response to Reviewers

We appreciate the Reviewers' comments, and the opportunity to respond. Below, we address each comment raised by Reviewer 3 (Reviewer 2 had no further comments, and Reviewer 1 reviewed our manuscript for [REDACTED]). Many of the issues raised concern design choices that we carefully considered during the course of planning the study. We recognize that there are limitations and tradeoffs with any choice of design, and indeed our choices of rating scale, conditioning procedure, and other design elements were made to address the limitations inherent in some previous design choices in the field based on 20 years of experience with dozens of similar studies. We thus particularly appreciate the opportunity to explain some of the tradeoffs involved and the merits of our choices here. Overall, our findings converge with those from previous studies that used different rating scales, variants of conditioning protocols, and design details, increasing our confidence that these design choices did not critically affect the results. However, we believe the present study provides particularly definitive evidence due to the large sample size, use of *a priori* neuromarkers and regions of interest, tests across two different kinds of pain (conditioned and unconditioned), and use of Bayes factors (made possible with our large sample size). In addition, as we explain below, rigorous positive controls (internal validation) give us confidence that our scales and imaging procedures were appropriately sensitive. Thank you for bringing these issues to light and promoting open discussion.

Reviewer #2 (Remarks to the Author)

My comment has been appropriately addressed. Excellent paper.

We are grateful for the reviewer's support and thank them again for their previous helpful comments.

Reviewer #3 (Remarks to the Author)

The study by Botvinik-Nezer et al. is an investigation to deeply investigate mechanisms of placebo using activation signatures that hold the potential to be more sensitive to activation than traditional massive univariate approaches. Furthermore this investigation seeks to better test the transfer of placebo conditioning from one modality (heat) to mechanical stimulation. The tremendous efforts of the authors to collect a huge (n=392) fMRI dataset to address this important question are certainly appreciated. However, numerous major issues with pain ratings, stimulus delivery, and the conditioning paradigm raise serious questions about their central conclusion that higher order cognitive/affective brain regions but not nociceptive processing regions mediate the placebo response.

Comment 1: The central conclusion is based on the observation that the NPS was not significantly related to the placebo response, while the SIIPS was. However, there is no direct statistical comparison of the NPS vs SIIPS signals to support this conclusion. Was the SIIPS more related to the placebo response than the NPS?

Response 1:

We used mixed effects models and Bayes factors to test the placebo effect on outcomes, including pain ratings, the two neuromarkers (NPS and SIIPS), and pre-registered ROIs. It is unusual to perform a statistical test to compare two outcomes that are measured on different scales, which is why we did not do so originally. However, the two outcomes can be compared based on their effect sizes. The effect size of placebo on the NPS was $d = 0.04$ in thermal pain and $d = -0.02$ in mechanical pain, while for SIIPS it was $d = 0.19$ in thermal and $d = 0.15$ in mechanical pain. In the revised manuscript, to directly compare the neuromarkers, we z-scored the scores of each neuromarker across conditions and participants (note that the between and within-participant effects within each neuromarker will not be affected) and tested the effect of the neuromarker (NPS / SIIPS) on the score in a mixed effects model. Placebo-induced reductions were significantly larger for the SIIPS than the NPS, in both modalities (thermal: $p = .030$, mechanical: $p = .006$).

We have now added this result to the manuscript:

Results - thermal:

“Finally, the placebo-induced reduction in neuromarker score was significantly larger for the SIIPS compared to the NPS (estimate = 0.100, SE = 0.046, $t(678.81) = 2.175$, $p = .030$, 95% CI = [0.010, 0.189]; note that the neuromarker scores were z-scored within each neuromarker and modality, to allow a direct comparison between these two outcomes that are measured on different scales, see Methods).” (pages 14-15)

Results - mechanical:

“Moreover, the placebo-induced reduction in neuromarker score was again significantly larger for the SIIPS compared to the NPS (estimate = 0.124, SE = 0.045, $t(609.62) = 2.766$, $p = .006$, 95% CI = [0.036, 0.211]).” (page 15)

Methods:

“To directly compare the placebo-induced reduction between the NPS and SIIPS, we first z-scored the scores of each neuromarker across conditions (placebo/control and stimulus intensity) and participants, such that neuromarkers could be represented on the same standardized scale but the within and between-subject effects within each neuromarker were not affected. Then, we computed the control - placebo difference for each neuromarker for each combination of participant, modality and stimulus level, and compared NPS vs. SIIPS with the following mixed-effects model

$brain_score \sim 1 + neuromarker + stimulus_level + (1 + neuromarker + stimulus_level | family_ID) + (neuromarker_monozygotic + stimulus_level_monozygotic + intercept_monozygotic - 1 | participant_ID) + (neuromarker_dizygotic + stimulus_level_dizygotic + intercept_dizygotic - 1 | participant_ID)$ ” (page 38)

Furthermore, the use of Bayes factors, enabled by our large sample size, allowed us to quantify the evidence in favor of the null vs. alternative hypothesis (existence or absence of a placebo effect on the NPS), and conclude with confidence that there are no effects of placebo on the NPS, but there are significant placebo effects on the SIIPS. These findings converge with those of other studies that use different conditioning procedures and rating scales (e.g., no placebo effects on the NPS in ^{1,2} and 17/20 studies in ³). However, here we provide formal evidence in favor of the null for the first time, and provide the first definitive, large-scale findings of placebo-induced reductions in the SIIPS across two kinds of pain.

Comment 2: Moreover, the incorporation of individual differences into the NPS revealed that reductions in perceived pain intensity were, in fact, related to reductions in the NPS and nociceptive ROI's, bringing the findings more into line with existing studies.

Response 2:

Thank you for highlighting this point. Indeed, placebo-induced effects on NPS responses and pain ratings were correlated both here and in the meta-analysis of Zunhammer et al. 2018 ³. Furthermore, group-level placebo effects were small in both studies (but significant in the meta-analysis, while null based on Bayes factors in our paper). Thus, our findings are largely consistent with prior literature. In the present manuscript, we noted that these correlations could indicate a reduction of NPS in the strongest placebo responders, which is in line with other studies (e.g., ³⁻⁶). Still, Bayes factors also strongly favored a null response for the average person, implying that if some participants show placebo effects, some must also show *reverse* placebo effects (e.g., ⁷).

Importantly, we also point out that random variation across individuals in skin site sensitivity, as well as physiological processes like habituation and sensitization, could result in correlations such as these. We carefully designed the study to minimize such effects, e.g., using a Control - Placebo - Placebo - Control design, which eliminates person x order (sensitization / habituation) interactions, but they still exist and could drive the correlations we discovered. We believe that we have discussed these issues transparently, and that open discussion of the various implications for placebo effects on nociceptive systems will help move the field forward.

We now elaborate more on this in the revised manuscript:

Results:

“In the current paper, we have focused primarily on the causal effects of placebo treatment in experimental settings on pain at the group level, as reported above. Nevertheless, we also tested the correlations between individual differences in placebo analgesia and neural placebo-induced changes, as was done in previous studies. Importantly, correlations in small samples (as in most previous placebo fMRI studies) are unreliable, and meta-analyses cannot address this issue because of heterogeneity across studies. Thus, this is one of the first studies which could adequately test these behavioral-brain correlations. Nevertheless, such correlations do not imply causal effects of placebo, and can be driven by other processes (see Discussion).” (page 21)

Discussion:

“Finally, this study provided a rare opportunity to examine individual differences in neural placebo effects in a sufficiently large sample to allow for stable correlation estimates and detection of small effects typical of between-person correlations^{8,9}. We found significant correlations between behavioral analgesia and larger reductions in both the NPS and SIIPS, as well as some individual regions, including the aMCC, dpIns, PAG and medial thalamus. This suggests that placebo treatment may cause reductions in the NPS in some individuals, as suggested by a previous person-level meta-analysis³. However, we cannot endorse this conclusion here because such correlations cannot provide strong evidence for causal effects. Here, as in previous literature, the strength of an individual’s behavioral analgesic effect is conflated with random variation in sensitivity on different skin sites and/or sensitization/habituation over time. For example, a participant whose placebo skin site is less sensitive than the control skin site is expected to report less pain for placebo compared to control irrespective of the placebo effect induced by the placebo manipulation, and also to have a lower NPS response for placebo compared to control. Because skin site sensitivity across participants is random, and skin site order was counterbalanced across participants, this random variation should not influence the main effect of placebo in the present study. It could, on the other hand, induce correlations between control minus placebo differences in pain ratings and neural responses, as we have found. This issue is common to virtually all clinical trials of treatment effects. Future studies could productively examine brain-behavioral correlations by selecting high and low placebo responders based on independent criteria (e.g., a separate session using different skin sites). Here, pre-scan expectation ratings could serve this goal, but expectancy ratings were not correlated with neural placebo-control differences, and were only associated with behavioral analgesia in thermal pain.” (page 27)

Comment 3: In studies of human pain mechanisms, obtaining accurate and precise reports of pain is critical. There are a number of well-described and well validated scales in the literature. Many are relatively continuous and have interval or ratio scale properties essential for use as a covariate of interest to interpret brain activity. In this study, the scales for rating pain are barely described and it appears that two separate scales are used. One is used in the “calibration” phase and appears to be a verbal

numerical scale. The other used in the main part of the study is some form of trackball controlled scale that appears to have verbal descriptors positioned at certain numbers and/or positions on the scale. The Bartoshuk reference is a review article that describes multiple types of scales and does not provide a description or validation of the scale that appears to be used during the fMRI portion of the study.

Response 3:

We are grateful for the opportunity to clarify and better describe the scale. Here, we address the issues raised with the scale. We address the “calibration” phase, which was actually a brief pre-experiment familiarization phase, in Response 12 below.

We chose to use the Labeled Magnitude Scale (LMS), which consists of quasi-logarithmically spaced perceptual verbal labels, precisely because it is commonly used and has been extensively validated. This scale was built off early psychophysical scale development^{10,11} and constructed by Green et al.^{12,13}. We have used such labeled magnitude scales in many of our studies since 2016, following over 10 years of experience with different pain rating scales, and they have been used in many previous pain studies in our lab and others (e.g.,¹⁴⁻²¹). The LMS has several advantages that led us to make this choice. First, it was designed and validated as a ratio scale, so that a two-fold increase in pain intensity is reflected in a rating that is twice as high on the scale²². Second, the scale’s descriptive labels (e.g., “weak”, “moderate”, “strong”) help participants describe their experience more precisely and consistently²³. Third, it allows to avoid ceiling effects on pain reports, by allowing room to increase ratings even for very painful stimulus intensities.

Beyond the extensive validation to the LMS scale in different modalities, we and others have validated that the LMS responds sensitively to manipulations of noxious stimulus intensity, providing internal validation^{18,24}. In addition, we have shown that LMS ratings correlate robustly with brain activity, in both thermal and mechanical pain, vicarious/observed pain, aversion to negative images, and aversive tastes, in a manner that is consistent across individuals and studies^{18,24}, providing external validation for the LMS in pain and beyond.

Finally, it is important to consider that our test of placebo effects is within-participant, rather than across groups, and is thus not very sensitive to the choice of scale (e.g., visual vs. labeled scales and choices of anchors)²⁵. Furthermore, the findings that the NPS was not affected by placebo and the SIIPS was affected – the main novel findings here – were based on brain activity, and did not depend on the behavioral pain ratings or choice of scale.

We have now revised the manuscript to clarify the scale that was used:

“We used a Labeled Magnitude Scale (LMS), consisting of quasi-logarithmically spaced perceptual verbal labels (0.014 = barely detectable, 0.061 = weak, 0.172 = moderate, 0.354 =

strong, 0.533 = very strong; ^{12,13}), which provides ratio properties ²⁶ and avoids ceiling effects on pain reports that are common with some narrow-range rating scales. Participants were presented with the intermediate scale labels during training, but they were removed when obtaining actual pain ratings during the experimental tasks, to minimize clustering around the labels ²⁷. After the scale's training, participants completed a familiarization task, to ensure their tolerance to the painful thermal stimuli. Stimuli in the familiarization task ranged between 45.5 - 48.5 °C, with a duration of 10 seconds each. Participants rated each stimulus verbally based on the same LMS scale. Participants were instructed that any sensation they would describe as pain should get an intensity rating above 0, and that the most intense pain they would normally tolerate should be rated between 0.5-0.6. ... Following each stimulus, participants rated the intensity (from "no pain" to "most pain imaginable") and unpleasantness (from "not at all" to "worst pain imaginable") of the stimulus." (pages 31-32)

Comment 4: Moreover, the instructions to distinguish intensity and unpleasantness are neither described nor validated.

Response 4:

We used standard instructions, following Price et al.'s pioneering work ²⁸⁻³¹. Distinguishing between intensity and unpleasantness is common in pain studies, with many examples from our lab and others (e.g., ^{16,18,29,32-40}). As we note at the beginning of the results section, the unpleasantness and intensity ratings were highly correlated (as was previously demonstrated in healthy participants ^{41,42}), and the choice to use the intensity ratings did not affect the conclusions. We now describe the scales and the instructions given to the participants in more detail (see excerpts above in Response 3).

Comment 5: The pain ratings of both the thermal and mechanical stimuli are quite low, with ratings of only .19/1 for heat and 0.13/1 for mechanical.

Response 5:

This is an important point to address, because the absolute values on the LMS can be misleading. Note that on the LMS scale, 0.06 = weak, 0.17 = moderate, 0.35 = strong, 0.53 = very strong and 1 = strongest imaginable. Even weak pain would be labeled as "painful" by participants, comparable to approximately 2 out of 10 on the numerical scales used in our previous work (e.g., ^{43,44}), and moderate pain (0.17) corresponds approximately to 5 out of 10. Furthermore, the strongest pain that participants will normally tolerate lies between 0.5 and 0.6, corresponding approximately to 8 out of 10. By design, participants rarely use the upper half of the scale, which is reserved for pains that would be intolerable in an experimental setting and thus unethical to deliver. This prevents ceiling effects in reported pain, which can be a problem with some other scales.

Two other points support the inference that the stimuli elicited adequate levels of pain. First, the stimuli used (46.5 - 47.5 degrees) were above the firing threshold for specific nociceptors⁴⁵. They were chosen to be painful to participants while tolerable to a broad population without causing participant dropout and sampling bias. Second, the LMS shows sensitivity to increasing levels of noxious heat, providing additional internal validation, and correlations with brain measures validated to be pain-related using different studies and methods (the NPS and SIIPS), providing additional external validation. The NPS in particular is insensitive to variations in stimulus intensity in the innocuous range¹, but was sensitive to stimulus intensity here.

Comment 6: This raises major questions about whether sufficient brain activation by these apparently weak stimuli was to detect a placebo reduction. This becomes particularly important when the main conclusion is that the nociceptive network was not modulated but the more cognitive/affective network was modulated by placebo. Could this just be a floor effect since the ratings were so low to begin with? This appears to contradict a relatively large body of placebo literature – including the very impressive meta-analysis by Zunhammer et al. (and some of the present authors) showing involvement of the medial and lateral thalamus, posterior insula, ACC, and other regions that have direct spino-thalamic input and are associated with nociceptive processing.

Response 6:

As reported in the text and shown in Figure 3b, the NPS was significantly activated in pain vs. rest in both modalities with large effect sizes (thermal $d = 1.11$; mechanical $d = 1.02$). These effect sizes are twice as large as typical positive findings in the Human Connectome Project data⁴⁶ (e.g., motor activity in individual motor cortex voxels) and comparable to the very largest such effects in other task domains. Moreover, there was a highly significant effect of the stimulus intensity on the NPS and SIIPS, in both modalities. This serves as a positive control, demonstrating that there was no floor effect and that the fMRI signal (and in particular the NPS and SIIPS) showed adequate sensitivity to the stimuli.

With regard to previous studies, the placebo effect on the NPS in Zunhammer et al., 2018³, was significant but very small across all 20 studies included in the meta-analysis ($g = 0.08$), and was found only in 3 out of the 20 individual studies. As for the placebo effect in individual regions reported in Zunhammer et al., 2021⁵, of the nociceptive pain brain regions, a main effect of placebo was only found in a specific part of the right insula, with a small effect size. There was no significant placebo vs. control treatment effect on other classical pain regions (including medial and lateral thalamus, posterior insula, and ACC), except for the middle cingulate cortex, even when the studies were modeled as a fixed rather than a random effect. This raises questions about whether there are placebo effects on the NPS, that are addressed in the present study using Bayes Factors. However, Zunhammer et al. did find significant correlations between the placebo effect

and the placebo-induced neural reductions in several pain regions, including the thalamus and insula. As discussed above, such correlations (individual differences) were also found in the present study, but they are not evidence of a causal effect of placebo on activity in these regions. The present study thus extends these previous findings.

We revised the discussion to more thoroughly compare our findings with previous studies and elaborate on the individual differences finding:

“Despite these substantial behavioral effects, fMRI analyses did not reveal significant reduction in the NPS—the most widely validated neuromarker of nociceptive pain to date—or nociception-related ROIs that were pre-registered based on previous studies. Bayes Factor analyses showed strong evidence in favor of null effects. Bayesian evidence for null effects was also found in individual subregions, precluding the possibility that the null effects resulted from a mix of positive and negative findings in different regions. In addition, in the novel test of transfer to mechanical pain, placebo treatment caused paradoxical activity increases in several of the regions most closely associated with pain processing, including aMCC, dpINS, and sensory thalamus (VPL/VPM; see below for discussion of these findings). The NPS has been shown to be sensitive to bottom-up stimulus intensity across multiple pain types^{3,47}, including sensitivity to both thermal and mechanical pain intensity in this study, and our sample size provided high power to detect small effects and provide strong Bayesian evidence in favor of the null. With $n = 392$ and $p < 0.05$, we have more than 80% power to detect “very small” effects of $d = 0.15$, and ~100% power to detect “small” effects of $d = 0.3$. The most recent meta-analysis revealed a significant placebo effect on the NPS, but this effect was very small ($g = 0.08$) and was found only in 3 out of 20 individual studies³. Moreover, of the nociceptive pain regions, a significant placebo effect was only found in a specific part of the insula, with a small effect size (and also the middle cingulate cortex when studies were modeled as a fixed rather than a random effect)⁵. Thus, together, these findings suggest that placebo analgesia is not driven by modulation of low-level nociceptive processes, at least for the average participant in experimental settings (for a discussion of individual differences see below).” (pages 24-25)

“Finally, this study provided a rare opportunity to examine individual differences in neural placebo effects in a sufficiently large sample to allow for stable correlation estimates and detection of small effects typical of between-person correlations^{8,9}. We found significant correlations between behavioral analgesia and larger reductions in both the NPS and SIIPS, as well as some individual regions, including the aMCC, dpIns, PAG and medial thalamus. This suggests that placebo treatment may cause reductions in the NPS and nociceptive brain regions in some individuals, as suggested by a previous person-level meta-analysis^{3,5}. However, we cannot endorse this conclusion here because such correlations cannot provide strong evidence for causal effects. Here, as in previous literature, the strength of an individual’s behavioral analgesic effect is conflated with random variation in sensitivity on different skin sites and/or

sensitization/habituation over time. For example, a participant whose placebo skin site is less sensitive than the control skin site is expected to report less pain for placebo compared to control irrespective of the placebo effect induced by the placebo manipulation, and also to have a lower NPS response for placebo compared to control. Because skin site sensitivity across participants is random, and skin site order was counterbalanced across participants, this random variation should not influence the main effect of placebo in the present study. It would, on the other hand, induce correlations between control minus placebo differences in pain ratings and neural responses, as we have found. This issue is common to virtually all clinical trials of treatment effects. Future studies could productively examine brain-behavioral correlations by selecting high and low placebo responders based on independent criteria (e.g., a separate session using different skin sites). Here, pre-scan expectation ratings could serve this goal, but expectancy ratings were not correlated with neural placebo-control differences, and were only associated with behavioral analgesia in thermal pain.” (page 27)

Comment 7: Also, the ratings in the text are lower than those in figure 2 although they are both supposedly means of the same data.

Response 7:

We checked the pain ratings reported in the text and those shown in Figure 2, and they are the same. The reported means are averaged across conditions (placebo / control) or stimulus intensities (low / middle / high), which may have confused the reviewer. We added a note in the text to clarify:

“As shown in Figure 2, thermal pain ratings increased with stimulus intensity (M [averaged across conditions] = 0.129, 0.148, and 0.191 for low, medium, and high intensity; Intensity effect: $\beta = 0.409$, $SE = 0.031$, $t_{(416.2)} = 13.32$, $p < .001$, 95% $CI = [0.348, 0.469]$) and were lower in the Placebo compared to the Control condition (Placebo $M = 0.129$, Control $M = 0.183$; Placebo effect: $\beta = -0.359$, $SE = 0.037$, $t_{(230.9)} = -9.73$, $p < .001$, 95% $CI = [-0.432, -0.286]$, $d = 0.53$).” (page 6)

Comment 8: The low ratings may arise from two factors. First, glabrous skin can be a great deal less sensitive to noxious stimuli than hairy skin – likely due to the absence of some populations of A-delta nociceptors. Thus, the noxious stimuli may have simply not been sufficiently intense to elicit much of a perceptual response and concomitant activation of nociceptive processing mechanisms. Second, the measurement properties of the pain scale are unknown based on the absence of any citations regarding the validation of this scale (or scales since two seemed to have been used). Is it possible that the positioning of the labels both drove ratings down and caused them to cluster in specific positions? The SEMs of the ratings are miniscule.

Response 8:

We are aware of this difference in skin sensitivity (and also discussed it in the limitations paragraph in the discussion), and we conducted extensive pilot studies to choose stimulus intensity parameters that were painful on these skin sites. As explained above, the LMS scale indeed lowers the ratings compared to linear scales, since a larger part of the scale is allocated for very strong stimuli. However, the ratings in the current study indicate that the stimuli were sufficiently painful for participants (see also responses 3 and 5 above). As for the labels, the intermediate labels were removed during the tasks, and thus did not drive clustering of ratings. As indicated in the figure caption, the error bars represent within-participant standard error of the mean, based on Morey, 2008⁴⁸, rather than between participant standard error (since the within-participant standard error better represents the mixed effects models that are used).

Comment 9: Another major concern of the experimental design is that there were two forms of placebo conditioning used – a symbolic and a classical conditioning. The symbolic conditioning paradigm is of significant concern given the results. This paradigm involved probe placement and then presentation of fake example ratings for the control vs placebo condition. This raises concerns that demand characteristics of this procedure would lead to significant biases of ratings to conform to those provided by the fake participants rather than to reflect what participants actually felt. Thus, was the differential activation in the cognitive/affective areas related to rating biases and/or the formulation of ratings rather than actual modulation of the experience of pain?

Response 9:

We chose the symbolic conditioning procedure because previous studies using symbolic conditioning actually produced larger effects on the NPS and areas related to nociceptive pain² (study 6),⁴⁹ than previous standard conditioning paradigms (e.g., those used in^{6,43}, and many papers since). Any procedure that informs people that they should experience more or less pain (i.e., virtually any placebo manipulation) and then asks them about that pain could be influenced by demand characteristics, which is an important reason to assess pain-related physiology. In previous studies, symbolic conditioning produced placebo effects in autonomic physiology as well as self-report, which is a brainstem-mediated process not subject to demand characteristics⁵⁰. In addition, the effect sizes we observed here for placebo effects on pain reports were close to those in Zunhammer et al., 2018 meta-analysis, $g = 0.66$ in the meta-analysis and $d = 0.53$ here, suggesting that there is not an extra demand characteristic here. The effects on the NPS were also similar, with $g = 0.08$ in Zunhammer et al., 2018 and $d = 0.04$ here. This suggests that the equivalent behavioral placebo effects are not due to an antinociceptive effect in one study versus a demand characteristic effect in the present study. Furthermore, pre-scan expectations were not significantly correlated with the SIIPS placebo-induced reductions, suggesting that the placebo effects on SIIPS do not represent effects of demand characteristics. In sum, the evidence suggests that symbolic

conditioning does not preferentially predispose to demand characteristics, and we are aware of no direct evidence for this possibility.

Since pain reports are the result of an evaluative decision, to question whether a treatment that affects pain reports is the result of “actual modulation of experience” is to question whether pain reports themselves represent “actual experience.” We are not aware of any research that has ever been able to show that a pain report is definitively not the result of an actual subjective experience. What we can show with this study is that (1) brain systems associated with nociceptive pain (e.g., the NPS) track self-reported pain but do not show placebo effects; and (2) brain systems associated with internal variables, including evaluation of pain and presumably decision-making about pain (e.g., the SIIPS) track self-reported pain and do show placebo effects. The future implications of which types of evaluative contributions to pain are reflected in the SIIPS, and whether they are trivial (“biases”) or consequential (long-term changes in pain decision-making and behavior) is an important question, but is beyond the scope of this study.

We added discussion on these issues:

“Sixth, like any study manipulating participants’ expectations, demand characteristics may drive participants to report less pain in the placebo condition. The size of the placebo effect on pain ratings was comparable to previous studies (e.g., $d = 0.53$ here and $g = 0.66$ in Zunhammer et al., 2018³), suggesting that there is not an extra demand characteristic effect in the present study. Furthermore, we revealed significant placebo effects on the SIIPS, and pre-scan expectations were not correlated with this effect across participants, suggesting that these neural effects do not represent effects of demand characteristics. Future studies are needed to examine which types of evaluative contributions to pain are reflected in the SIIPS, and whether they are trivial (“biases”) or consequential (long-term changes in pain decision-making and behavior).” (page 29)

Comment 10: Moreover, the use of two placebo paradigms in series seems likely to further complicate interpretation of the results and would likely be largely unnecessary given the demonstrated strength of the classical conditioning paradigm. Would the symbolic conditioning paradigm engage different mechanisms than the classical conditioning paradigm? Would some participants use one vs the other, or even both together?

Response 10:

As discussed above, we chose to combine symbolic conditioning with reinforcement because it was previously shown to have stronger effects on the NPS and related brain responses. In addition, current theoretical models suggest that suggestion, vicarious experience, conditioning, and other manipulations work together to strengthen expectations (e.g.,⁵¹) and related neural predictions of reduced pain under placebo treatment (i.e., predictive processing,⁵²), and it has been recently

shown with a randomized, controlled trial that a combination of placebo paradigms indeed induces the strongest placebo effect⁵³. Thus, we combined placebo suggestion, symbolic conditioning, and classical conditioning, giving the study the maximal chances of showing behavioral and neural placebo effects. Moreover, symbolic conditioning allowed us to increase the number of learning trials (without increasing the number of thermal stimuli), which is associated with stronger placebo responses in previous work (e.g.,⁵⁴). We therefore have no reason to believe that reinforcing suggestions first with symbolic pain outcomes (modeled after^{2 (study 6),50}) and then with actual noxious outcomes would weaken the effect. Indeed, our behavioral results suggest that it did not. We also saw no evidence in the data that different participants were using different strategies or susceptible to different parts of the manipulation, and speculating about such tradeoffs is not theoretically or empirically justified. Future studies could systematically compare different components of the paradigm and test whether different brain systems are involved in the induced placebo effect (as was done in the past for example for expectation effects induced via social cues vs. conditioning⁵⁵). We revised the methods and discussion sections to provide additional details about the different components of the placebo induction paradigm, the motivation for including them, and the potential implications.

Methods:

“To strengthen expectations of pain relief from the placebo treatment cream, participants were subjected to two conditioning paradigms⁵³. Importantly, both conditioning paradigms were based on thermal, not mechanical, stimuli. First a “symbolic” conditioning paradigm was administered. This paradigm was similar to previous studies^{50,56 (study 6)}: An inert thermode was placed on the control site and the participants completed a trial sequence mimicking a thermal stimulation sequence, except instead of thermal stimuli they were shown ratings they were told come from prior participants. These ratings were systematically high for 16 stimuli. The thermode was then moved to the prodicaine treated site and the procedure was repeated for 32 stimuli. This time the ratings were systematically low. Finally the thermode was moved back to the control site and 16 additional ratings were once again systematically high. The thermode was placed on the proximal phalanges. Second, a classical condition paradigm was administered. This paradigm was identical to the symbolic conditioning paradigm, except instead of being shown ratings of other participants, participants were subjected to noxious thermal stimulation of the proximal phalanges. They were told that stimuli were all of the same intensity, but the intensity was surreptitiously lowered by 3.5 °C when stimulating the prodicaine treated site compared to the control treated site (44 and 44.5 °C for the placebo cream and 47.5 and 48 °C for the control cream). Following each stimulus, participants rated the intensity (from “no pain” to “most pain imaginable”) and unpleasantness (from “not at all” to “worst pain imaginable”) of the stimulus. After the conditioning task, participants rated their expectations regarding the Prodicaine efficacy in the next task, on a linear scale between 0 [not at all] to 100 [most effective].” (page 32)

Discussion:

“Fourth, different placebo-induction protocols or analysis pipelines may lead to different findings⁵⁷, for example with regard to the effect of placebo on nociceptive processes. Here, the placebo induction combined several components in order to maximize placebo effects⁵³ (e.g., suggestion, conceptual conditioning^{50,56 (study 6)} and classical conditioning)^{51,52}, which yielded strong behavioral effects, in addition to an unconditioned transfer condition (which produced similarly large behavioral and neural effects). More studies are needed in order to systematically compare different components of the paradigm and test whether different brain systems are involved in the induced placebo effect (see for example^{53,55}).” (page 28)

Comment 11: Another concern regarding the lack of modulation of nociceptive processing regions arises from the relatively minimal number of trials involving noxious stimulation. There were 6 stimuli (low, med, high X thermal vs mechanical) and there were 8 stimuli per run, with two control runs and two placebo runs. Thus, most stimuli were only repeated twice during each control and each placebo condition. Also, other temperatures are mentioned during the conditioning procedure. Were these also used in the fMRI portion? Detecting change in this very complicated design (3x2x2) with perceptually weak stimuli and minimal repeats of individual stimuli seems challenging.

Response 11:

First, as noted above, based on the literature as well as the observed ratings in the present study, the stimuli were not perceptually weak. Regarding the number of trials, there were 8 trials in total for each combination of modality (thermal / mechanical) and condition (control / placebo), and each noxious stimulus lasted 10 seconds. We focused on main placebo effects across all three levels of stimulus intensity within each modality (and there were no significant interaction effects between the stimulus intensity and the condition across behavioral and brain measures). Some previous studies of placebo effects (e.g.,⁵⁸⁻⁶⁴) were similarly conducted with 10 trials or fewer per condition. Moreover, the internal validation analyses in the present study show ample sensitivity with this number of trials to detect stimulus intensity effects, as well as placebo effects on the SIIPS and several ROIs. With our sample size of $N = 392$ and $p < .05$ we have more than 80% power to detect “very small” effects of $d = 0.15$, and ~100% power to detect “small” effects of $d = 0.3$ or larger effects. Thus, we had adequate power to detect placebo effects on the NPS and other nociceptive pain-related regions should even small effects have been present. In sum, we can assure the editors and reviewers with confidence that the number of trials was adequate and not an explanation for our findings of placebo effects on some brain measures but not others.

Comment 12: The “calibration” procedure is hopefully misnamed. For most researchers in the field, “calibration” would mean that stimuli are individually adjusted for each participant based on their ratings. This process introduces huge complexity in since different stimuli are used in different individuals, different populations of primary afferents are likely activated, with unknown impact on brain activity or the ability of pain

to be modulated. In contrast, “training” or “familiarization” would imply that participants were exposed to the stimuli to see if they could tolerate them and to gain experience providing pain ratings. Unfortunately, it seems that different pain scales were used during familiarization and fMRI, so there was no benefit of training with scale usage. Also the site of “calibration” was not defined.

Response 12:

This task was indeed mislabeled, and should be named “familiarization.” We apologize for this error, and thank the reviewer for drawing our attention to it. This task was used to familiarize participants with the thermal stimuli and ensure that they are tolerable. The skin site that was used for this task was the right index finger. The rating scale was trained in a separate stage, performed prior to the familiarization task. The stimuli in the present experiment were not calibrated for each participant, but rather fixed temperatures were used in each task as explained in the Methods section (and in our response to comment 5). We have now corrected the name of the task and revised the methods section to provide a detailed description of all tasks.

Methods:

“We used a Labeled Magnitude Scale (LMS), consisting of quasi-logarithmically spaced perceptual verbal labels (0.014 = barely detectable, 0.061 = weak, 0.172 = moderate, 0.354 = strong, 0.533 = very strong; ^{12,13}), which provides ratio properties ²⁶). and avoids ceiling effects on pain reports that are common with some narrow-range rating scales. Participants were presented with the intermediate scale labels during training, but they were removed when obtaining actual pain ratings during the experimental tasks, to minimize clustering around the labels ²⁷.

After the scale’s training, participants completed a familiarization task, to ensure their tolerance to the painful thermal stimuli. Stimuli in the familiarization task ranged between 45.5 - 48.5 °C, with a duration of 10 seconds each. Participants rated each stimulus verbally based on the same LMS scale. Participants were instructed that any sensation they would describe as pain should get an intensity rating above 0, and that the most intense pain they would normally tolerate should be rated between 0.5-0.6. Participants who were unable to tolerate the stimuli were excluded from further participation. The thermal stimuli were delivered with a 16 X 16 mm surface thermode (PATHWAY ATS; Medoc, Inc, Israel). The skin site used for the familiarization task was the right index finger.” (pages 32-33)

References

1. Wager, T. D. *et al.* An fMRI-based neurologic signature of physical pain. *N. Engl. J. Med.* **368**, 1388–1397 (2013).
2. Woo, C.-W. *et al.* Quantifying cerebral contributions to pain beyond nociception. *Nature Communications* vol. 8 Preprint at <https://doi.org/10.1038/ncomms14211> (2017).
3. Zunhammer, M., Bingel, U., Wager, T. D. & Placebo Imaging Consortium. Placebo Effects on the Neurologic Pain Signature: A Meta-analysis of Individual Participant Functional Magnetic Resonance Imaging Data. *JAMA Neurol.* **75**, 1321–1330 (2018).
4. Tinnermann, A., Geuter, S., Sprenger, C., Finsterbusch, J. & Büchel, C. Interactions between brain and spinal cord mediate value effects in nocebo hyperalgesia. *Science* **358**, 105–108 (2017).
5. Zunhammer, M., Spisák, T., Wager, T. D., Bingel, U. & Placebo Imaging Consortium. Meta-analysis of neural systems underlying placebo analgesia from individual participant fMRI data. *Nat. Commun.* **12**, 1391 (2021).
6. Eippert, F. *et al.* Activation of the opioidergic descending pain control system underlies placebo analgesia. *Neuron* **63**, 533–543 (2009).
7. Scott, D. J. *et al.* Placebo and nocebo effects are defined by opposite opioid and dopaminergic responses. *Arch. Gen. Psychiatry* **65**, 220–231 (2008).
8. Marek, S. *et al.* Reproducible brain-wide association studies require thousands of individuals. *Nature* **603**, 654–660 (2022).
9. Spisak, T., Bingel, U. & Wager, T. Replicable multivariate BWAS with moderate sample sizes. *bioRxiv* 2022.06.22.497072 (2022) doi:10.1101/2022.06.22.497072.
10. Gracely, R. H., McGrath, P. & Dubner, R. Ratio scales of sensory and affective verbal pain descriptors. *Pain* **5**, 5–18 (1978).
11. Gav, B. A category scale with ratio properties for intermodal and interindividual comparisons. *Psychophysical judgment and the process of perception* 25–34 (1982).

12. Green, B. G., Shaffer, G. S. & Gilmore, M. M. Derivation and evaluation of a semantic scale of oral sensation magnitude with apparent ratio properties. *Chem. Senses* **18**, 683–702 (1993).
13. Green, B. G. *et al.* Evaluating the 'Labeled Magnitude Scale' for measuring sensations of taste and smell. *Chem. Senses* **21**, 323–334 (1996).
14. Goldstein, P., Losin, E. A. R., Anderson, S. R., Schelkun, V. R. & Wager, T. D. Clinician-Patient Movement Synchrony Mediates Social Group Effects on Interpersonal Trust and Perceived Pain. *J. Pain* **21**, 1160–1174 (2020).
15. González-Fernández, M. *et al.* Moving beyond the limitations of the visual analog scale for measuring pain: novel use of the general labeled magnitude scale in a clinical setting. *Am. J. Phys. Med. Rehabil.* **93**, 75–81 (2014).
16. Matthewson, G. M., Woo, C.-W., Reddan, M. C. & Wager, T. D. Cognitive self-regulation influences pain-related physiology. *Pain* **160**, 2338–2349 (2019).
17. Losin, E. A. R., Anderson, S. R. & Wager, T. D. Feelings of Clinician-Patient Similarity and Trust Influence Pain: Evidence From Simulated Clinical Interactions. *J. Pain* **18**, 787–799 (2017).
18. Čeko, M., Kragel, P. A., Woo, C.-W., López-Solà, M. & Wager, T. D. Common and stimulus-type-specific brain representations of negative affect. *Nat. Neurosci.* **25**, 760–770 (2022).
19. Geha, P. *et al.* Pharmacotherapy for Pain in a Family With Inherited Erythromelalgia Guided by Genomic Analysis and Functional Profiling. *JAMA Neurol.* **73**, 659–667 (2016).
20. Lee, D. H., Lee, S. & Woo, C.-W. Decoding Pain: Uncovering the Factors that Affect Performance of Neuroimaging-Based Pain Models. *bioRxiv* 2023.12.22.573021 (2023) doi:10.1101/2023.12.22.573021.
21. Huang, T. *et al.* Identifying the pathways required for coping behaviours associated with sustained pain. *Nature* **565**, 86–90 (2019).

22. Bartoshuk, L. M., Fast, K. & Snyder, D. J. Differences in Our Sensory Worlds: Invalid Comparisons With Labeled Scales. *Curr. Dir. Psychol. Sci.* **14**, 122–125 (2005).
23. Bartoshuk, L. M. *et al.* Labeled scales (e.g., category, Likert, VAS) and invalid across-group comparisons: what we have learned from genetic variation in taste. *Food Qual. Prefer.* **14**, 125–138 (2003).
24. Krishnan, A. *et al.* Somatic and vicarious pain are represented by dissociable multivariate brain patterns. *Elife* **5**, (2016).
25. Bartoshuk, L. M. *et al.* Valid across-group comparisons with labeled scales: the gLMS versus magnitude matching. *Physiol. Behav.* **82**, 109–114 (2004).
26. Bartoshuk, L. M. Comparing sensory experiences across individuals: recent psychophysical advances illuminate genetic variation in taste perception. *Chem. Senses* **25**, 447–460 (2000).
27. Hayes, J. E., Allen, A. L. & Bennett, S. M. Direct comparison of the generalized Visual Analog Scale (gVAS) and general Labeled Magnitude Scale (gLMS). *Food Qual. Prefer.* **28**, 36–44 (2013).
28. Price, D. D. Psychological and neural mechanisms of the affective dimension of pain. *Science* **288**, 1769–1772 (2000).
29. Price, D. D., McHaffie, J. G. & Larson, M. A. Spatial summation of heat-induced pain: influence of stimulus area and spatial separation of stimuli on perceived pain sensation intensity and unpleasantness. *J. Neurophysiol.* **62**, 1270–1279 (1989).
30. Price, D. D., Harkins, S. W. & Baker, C. Sensory-affective relationships among different types of clinical and experimental pain. *Pain* **28**, 297–307 (1987).
31. Price, D. D. & Harkins, S. W. The affective-motivational dimension of pain A two-stage model. *APS Journal* **1**, 229–239 (1992).
32. Rainville, P. Brain mechanisms of pain affect and pain modulation. *Curr. Opin. Neurobiol.* **12**, 195–204 (2002).

33. Rainville, P., Carrier, B., Hofbauer, R. K., Bushnell, C. M. & Duncan, G. H. Dissociation of sensory and affective dimensions of pain using hypnotic modulation. *Pain* **82**, 159–171 (1999).
34. Rainville, P., Duncan, G. H., Price, D. D., Carrier, B. & Bushnell, M. C. Pain affect encoded in human anterior cingulate but not somatosensory cortex. *Science* **277**, 968–971 (1997).
35. Hofbauer, R. K., Rainville, P., Duncan, G. H. & Bushnell, M. C. Cortical representation of the sensory dimension of pain. *J. Neurophysiol.* **86**, 402–411 (2001).
36. Woo, C.-W., Roy, M., Buhle, J. T. & Wager, T. D. Distinct brain systems mediate the effects of nociceptive input and self-regulation on pain. *PLoS Biol.* **13**, e1002036 (2015).
37. Losin, E. A. R. *et al.* Neural and sociocultural mediators of ethnic differences in pain. *Nat Hum Behav* **4**, 517–530 (2020).
38. Lee, S. A. *et al.* Brain Representations of Affective Valence and Intensity in Sustained Pleasure and Pain. *bioRxiv* 2023.06.08.544230 (2023) doi:10.1101/2023.06.08.544230.
39. Dunckley, P. *et al.* Cortical processing of visceral and somatic stimulation: differentiating pain intensity from unpleasantness. *Neuroscience* **133**, 533–542 (2005).
40. López-Solà, M. *et al.* Towards a neurophysiological signature for fibromyalgia. *Pain* **158**, 34–47 (2017).
41. Chapman, C. R. *et al.* Sensory and affective dimensions of phasic pain are indistinguishable in the self-report and psychophysiology of normal laboratory subjects. *J. Pain* **2**, 279–294 (2001).
42. Coghill, R. C., Gilron, I. & Iadarola, M. J. Hemispheric lateralization of somatosensory processing. *J. Neurophysiol.* **85**, 2602–2612 (2001).
43. Wager, T. D. *et al.* Placebo-induced changes in fMRI in the anticipation and experience of pain. *Science* **303**, 1162–1167 (2004).
44. Atlas, L. Y., Bolger, N., Lindquist, M. A. & Wager, T. D. Brain mediators of predictive cue effects on perceived pain. *J. Neurosci.* **30**, 12964–12977 (2010).

45. Churyukanov, M., Plaghki, L., Legrain, V. & Mouraux, A. Thermal detection thresholds of A δ - and C-fibre afferents activated by brief CO₂ laser pulses applied onto the human hairy skin. *PLoS One* **7**, e35817 (2012).
46. Poldrack, R. A. *et al.* Scanning the horizon: towards transparent and reproducible neuroimaging research. *Nat. Rev. Neurosci.* **18**, 115–126 (2017).
47. Han, X. *et al.* Effect sizes and test-retest reliability of the fMRI-based neurologic pain signature. *Neuroimage* **247**, 118844 (2022).
48. Morey, R. D. Confidence Intervals from Normalized Data: A correction to Cousineau (2005). *Tutor. Quant. Methods Psychol.* **4**, 61–64 (2008).
49. Jepma, M., Koban, L., van Doorn, J., Jones, M. & Wager, T. D. Behavioural and neural evidence for self-reinforcing expectancy effects on pain. *Nat Hum Behav* **2**, 838–855 (2018).
50. Jepma, M. & Wager, T. D. Conceptual Conditioning: Mechanisms Mediating Conditioning Effects on Pain. *Psychol. Sci.* **26**, 1728–1739 (2015).
51. Kirsch, I. Response expectancy theory and application: A decennial review. *Appl. Prev. Psychol.* **6**, 69–79 (1997).
52. Büchel, C., Geuter, S., Sprenger, C. & Eippert, F. Placebo analgesia: a predictive coding perspective. *Neuron* **81**, 1223–1239 (2014).
53. van Lennep, J. H. P. A. *et al.* The Optimal Learning Cocktail for Placebo Analgesia: A Randomized Controlled Trial Comparing Individual and Combined Techniques. *J. Pain* **24**, 2240–2256 (2023).
54. Colloca, L., Petrovic, P., Wager, T. D., Ingvar, M. & Benedetti, F. How the number of learning trials affects placebo and nocebo responses. *Pain* **151**, 430–439 (2010).
55. Koban, L., Jepma, M., López-Solà, M. & Wager, T. D. Different brain networks mediate the effects of social and conditioned expectations on pain. *Nat. Commun.* **10**, 4096 (2019).
56. Woo, C.-W. *et al.* Quantifying cerebral contributions to pain beyond nociception. *Nat.*

- Commun.* **8**, 14211 (2017).
57. Botvinik-Nezer, R. *et al.* Variability in the analysis of a single neuroimaging dataset by many teams. *Nature* **582**, 84–88 (2020).
58. Craggs, J. G., Price, D. D., Perlstein, W. M., Verne, N. G. & Robinson, M. E. The dynamic mechanisms of placebo induced analgesia: Evidence of sustained and transient regional involvement. *Pain* **139**, 660–669 (2008).
59. Koban, L., Kross, E., Woo, C.-W., Ruzic, L. & Wager, T. D. Frontal-Brainstem Pathways Mediating Placebo Effects on Social Rejection. *J. Neurosci.* **37**, 3621–3631 (2017).
60. Choi, J. C. *et al.* Placebo effects on analgesia related to testosterone and premotor activation. *Neuroreport* **22**, 419–423 (2011).
61. Bingel, U. *et al.* The effect of treatment expectation on drug efficacy: imaging the analgesic benefit of the opioid remifentanyl. *Sci. Transl. Med.* **3**, 70ra14 (2011).
62. Ellingsen, D.-M. *et al.* Placebo improves pleasure and pain through opposite modulation of sensory processing. *Proc. Natl. Acad. Sci. U. S. A.* **110**, 17993–17998 (2013).
63. Lui, F. *et al.* Neural bases of conditioned placebo analgesia. *Pain* **151**, 816–824 (2010).
64. Theysohn, N. *et al.* Are there sex differences in placebo analgesia during visceral pain processing? A fMRI study in healthy subjects. *Neurogastroenterol. Motil.* **26**, 1743–1753 (2014).

REVIEWERS' COMMENTS

Reviewer #3 (Remarks to the Author):

The authors have been highly responsive to my pointed critique and have done an outstanding job clarifying critical issues related to pain assessment. The new analyses comparing the NPS responses to SIIPS responses have considerably strengthened the conclusions. The authors provide a more comprehensive and nuanced discussion about the relation of the present findings of no to minimal placebo relationship with the NPS and places it within the context of the extant literature.

I still have some remaining concerns about the different ways of obtaining ratings from the LMS. Verbal vs. trackball, labeled vs unlabeled etc. These may have some effects on the ratings that were obtained. I think the impact of this in the present study would be limited. In future studies, keeping the scale and the reporting identical across all phases of the experiment would minimize these concerns.

REVIEWERS' COMMENTS

Reviewer #3 (Remarks to the Author):

The authors have been highly responsive to my pointed critique and have done an outstanding job clarifying critical issues related to pain assessment. The new analyses comparing the NPS responses to SIIPS responses have considerably strengthened the conclusions. The authors provide a more comprehensive and nuanced discussion about the relation of the present findings of no to minimal placebo relationship with the NPS and places it within the context of the extant literature.

I still have some remaining concerns about the different ways of obtaining ratings from the LMS. Verbal vs. trackball, labeled vs unlabeled etc. These may have some effects on the ratings that were obtained. I think the impact of this in the present study would be limited. In future studies, keeping the scale and the reporting identical across all phases of the experiment would minimize these concerns.

Response:

We thank the reviewer for the constructive and helpful feedback. Their feedback substantially improved the manuscript by helping us clarify important methodological aspects of our study as well as its contribution to the literature.